# Physics-Informed Audio-Geometry-Grid Representation Learning for Universal Sound Source Localization

**Min-Sang Baek, Gyeong-Su Kim, Donghyun Kim & Joon-Hyuk Chang***
Department of Electronic Engineering, Hanyang University, Seoul, Republic of Korea
`{kng643,rudtn9862,utahboy3502,jchang}@hanyang.ac.kr`

## Abstract

Sound source localization (SSL) is a fundamental task in spatial audio understanding, yet most deep neural network-based methods are constrained by fixed array geometries and predefined directional grids, limiting generalizability and scalability. To address these issues, we propose *audio-geometry-grid representation learning* (AGG-RL), a novel framework that jointly learns audio-geometry and grid representations in a shared latent space, enabling both geometry-invariant and grid-flexible SSL. Moreover, to enhance generalizability and interpretability, we introduce two physics-informed components: a *learnable non-uniform discrete Fourier transform* (LNuDFT), which optimizes the dense allocation of frequency bins in a non-uniform manner to emphasize informative phase regions, and a *relative microphone positional encoding* (rMPE), which encodes relative microphone coordinates in accordance with the nature of inter-channel time differences. Experiments on synthetic and real datasets demonstrate that AGG-RL achieves superior performance, particularly under unseen conditions. The results highlight the potential of representation learning with physics-informed design towards a universal solution for spatial acoustic scene understanding across diverse scenarios.

## 1 Introduction

With the growing interest in learning and reasoning about 3D space in machine learning (Mildenhall et al., 2021; Zhang et al., 2025; Xie et al., 2025), multichannel audio has emerged as a promising modality for capturing spatial cues and enabling spatially aware generation (Ochiai et al., 2017; Simeoni et al., 2019; Richard et al., 2021; Leng et al., 2022; Luo et al., 2022; Huang et al., 2023; Zheng et al., 2024; Sun et al., 2025; Brunetto et al., 2025; Li et al., 2025a; Liang et al., 2025). A central task in this field is *sound source localization* (SSL), which estimates the direction-of-arrival (DOA) of sound sources. SSL is not only a fundamental building block in spatial audio processing but also synergizes with applications such as robotics (Gan et al., 2020; Wang et al., 2021; Do et al., 2022; Wang et al., 2024a), autonomous vehicles (Furletov et al., 2021; Jeon et al., 2025), drones (Wang & Cavallaro, 2022; Chevtchenko et al., 2025), and immersive AR/VR systems and smart devices (Rajguru et al., 2022; Gupta et al., 2022; Lin et al., 2024; Yang et al., 2025).

Traditionally, SSL has relied on classical signal processing methods based on the time difference-of-arrival (TDOA) between microphones in a microphone array (MA) (Benesty et al., 2008), such as GCC-PHAT (Knapp & Carter, 1976), MUSIC (Schmidt, 1986), and SRP-PHAT (Dibiase, 2000), which exploit inter-channel phase differences (IPDs) in the frequency domain. The advent of deep neural networks (DNNs) has enabled models to learn robust audio representations, often surpassing classical methods (Grumiaux et al., 2022; Chakrabarty & Habets, 2019; Nguyen et al., 2020; Yang et al., 2021; Diaz-Guerra et al., 2021a; He et al., 2021; Baek et al., 2023; Zhang et al., 2024; Battula et al., 2025; Li et al., 2025b). However, most existing DNN-based methods remain critically limited: they are highly dependent on specific MA geometries and predefined DOA grids.

---

*Corresponding author.

To address these limitations, geometry-invariant SSL methods (Schwartz et al., 2023; Grinstein et al., 2024; Wang et al., 2024b; Baek et al., 2025) and grid-flexible approaches (Yang et al., 2021; Wang et al., 2024b) have been proposed. Although these methods alleviate some constraints, they still fall short of achieving robust SSL across arbitrary geometries and grids. Therefore, we propose *audio-geometry-grid representation learning* (AGG-RL) for robust, universal SSL. Inspired by representation learning (Bengio et al., 2013; Koyama et al., 2024a; Shi et al., 2022; Dimitriadis et al., 2023; Zhu et al., 2024; Kim et al., 2025), AGG-RL learns two types of representations: audio-geometry representations (AGRs) and grid representations (GRs). AGRs are derived from audio and MA geometry, while GRs encode candidate DOA grids. Their similarity produces a probabilistic spatial spectrum, which is trained with soft labels that represent relationships among neighboring grid points. This enables the model to capture audio-geometry-grid relationships, facilitating universal SSL on arbitrary grids and geometries without retraining.

Furthermore, to enhance generalization, we introduce two components inspired by physics-informed DNNs (Raissi et al., 2019; Karniadakis et al., 2021; Gabrielli et al., 2018; Koyama et al., 2024b; Ribeiro et al., 2024; Miotello et al., 2024; Karakonstantis et al., 2024; Damiano et al., 2025): a *learnable non-uniform discrete Fourier transform* (LNuDFT) and a *relative microphone positional encoding* (rMPE). For the proposed LNuDFT, the gap between adjacent frequency bins is modeled as a learnable parameter, allowing the DNN to densely allocate bins in critical frequency regions that convey physically informative phase cues. In addition, based on absolute MPE (aMPE) (Baek et al., 2025), rMPE is introduced to embed microphone coordinates in a relative fashion—consistent with the nature of TDOA, which depends on relative rather than absolute coordinates. Both components incorporate physics-based structural assumptions into the feature extraction process while still allowing adaptivity through training. This yields physics-informed inductive biases that guide learning toward acoustically meaningful representations.

Experiments on both synthetic and real datasets (Löllmann et al., 2018) demonstrate the effectiveness of the proposed method, showing strong generalization to unseen environments in terms of angular error and accuracy. Moreover, our framework supports flexible grid selection without retraining, enhancing adaptability across diverse scenarios. These results highlight AGG-RL, augmented with LNuDFT and rMPE, as a promising step toward 3D acoustic scene understanding.

## 2 BACKGROUND

### 2.1 TIME DIFFERENCE-OF-ARRIVAL AND INTER-CHANNEL PHASE DIFFERENCE

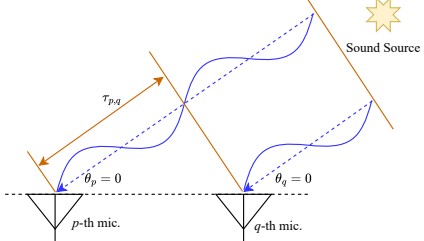

(a) Sound received by the microphone array.

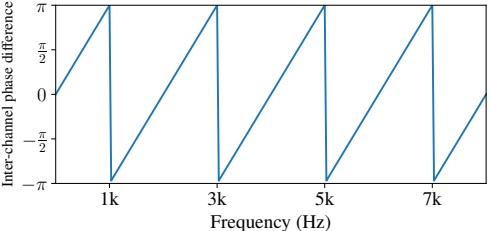

(b) IPD at $\tau = 0.5$ ms, showing phase wrapping and aliasing.

Figure 1: Illustration of (a) signals received by a microphone array and (b) IPD in the frequency domain, where phase wrapping can cause spatial aliasing.

Modeling a sound wave as a plane wave in the far-field situation, when the wave reaches spatially separated microphones at different times, a TDOA arises, as shown in Fig. 1(a). In the frequency domain, the TDOA $\tau_{p,q}$ between channels $p$ and $q$ is related to the IPD $\Delta\theta$ at frequency $f$ as follows:

$$e^{j(\theta_p - \theta_q)} = e^{j\Delta\theta} = e^{j(2\pi f \tau_{p,q})} \iff \tau_{p,q} = \frac{\Delta\theta}{2\pi f}, \tag{1}$$

where $j$ is the imaginary unit, and $\theta_p$ and $\theta_q$ denote the phases at channels $p$ and $q$, respectively. At higher frequencies, phase varies more rapidly, enabling finer resolution of the TDOA. However, since phase is wrapped within $[-\pi, \pi)$, the same IPD value may correspond to multiple TDOAs,

resulting in spatial aliasing, as illustrated in Fig. 1(b). The aliasing condition depends on the microphone spacing $r$ and the maximum unambiguous frequency $f_{\max}$:

$$f \leq f_{\max} = \frac{v}{2r} \iff r \leq \frac{\lambda}{2}, \quad \lambda = \frac{v}{f}, \tag{2}$$

where $v$ is the speed of sound, and $\lambda$ is the wavelength corresponding to $f$. Thus, aliasing can be avoided either by reducing the microphone spacing $r$ below half the wavelength of $f_{\max}$ or by limiting the frequency below $f_{\max}$. In summary, lower frequencies are alias-free but provide coarse TDOA resolution, while higher frequencies yield finer resolution at the cost of potential aliasing. As MA geometries vary widely across real-world applications, accounting for this trade-off is crucial for geometry-invariant SSL.

## 2.2 OUTPUT REPRESENTATION PARADIGMS IN DNN-BASED SSL

The outputs of DNN-based SSL can be broadly categorized into regression and classification approaches. Regression methods directly predict source locations in 3D coordinates (Diaz-Guerra et al., 2021a; Grinstein et al., 2024). They offer theoretically infinite resolution, but suffer from limited interpretability, since they predict only coordinates rather than a spatial spectrum indicating source likelihood across the entire space. Furthermore, their architectures are typically constrained by the maximum number of sources. Classification-based methods discretize the 3D space into predefined DOA grids (Schwartz et al., 2023; Baek et al., 2025). This yields interpretable spatial spectra without being tied to the number of sources, but performance is bounded by grid resolution, and retraining is required when adopting new grids. To balance these trade-offs, template matching (Yang et al., 2021; Wang et al., 2024b) has been explored. This approach enables SSL on arbitrary grids without retraining by comparing outputs with IPD templates from candidate DOAs. However, as the model is optimized for IPD estimation rather than direct DOA prediction, performance can degrade. Moreover, template matching requires pairwise outputs for each microphone pair and a predefined number of sources, leading to increased computational cost. These limitations motivate the proposed AGG-RL framework, which enables flexible-grid SSL with direct candidate DOA prediction, without relying on handcrafted templates or pairwise computations.

## 3 PROPOSED FRAMEWORK

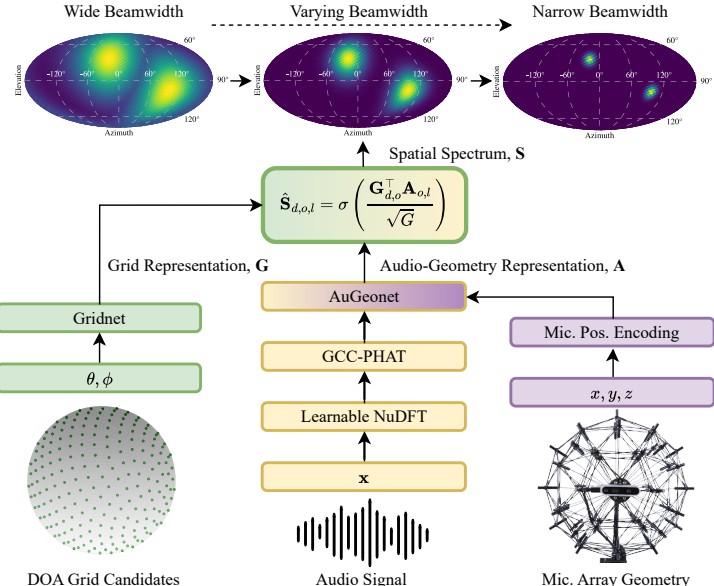

Figure 2: Overview of the proposed framework. The model takes audio signals, microphone array geometries, and candidate DOA grids as inputs, and outputs spatial spectra over the grids. Multiple oracle spatial spectra with different beamwidth parameters are also shown. The microphone array shown (Sphere48 AC Pro by gfai tech GmbH) is reproduced with permission.

To achieve grid-flexible and geometry-invariant SSL, we propose AGG-RL, illustrated in Fig. 2. The framework consists of two networks: the audio-geometry representation network (AuGeonet), $\mathcal{A}(\cdot)$, and the grid representation network (Gridnet), $\mathcal{G}(\cdot)$. AuGeonet integrates two physics-informed components—LNuDFT and rMPE—to extract robust AGRs, while Gridnet produces GRs from candidate DOA grids. AGRs and GRs are projected into a shared latent space, and their similarity is measured by an inner product, where larger values indicate higher likelihood of source presence. The models are trained using soft-labeled oracle spatial spectra (shown at the top of Fig. 2) as supervision, enabling effective learning of audio-geometry-grid relationships.

## 3.1 LEARNABLE NON-UNIFORM DISCRETE FOURIER TRANSFORM

In contrast to the standard DFT, which uses uniformly spaced frequency bins, the NuDFT (Bagchi & Mitra, 1996a;b; 1999) employs non-uniform frequency bin allocations that can be optimal for specific applications (Wei & Yang, 2022; Wen & Houlihan, 2023; de Haan et al., 2002; Chang, 2005; Muralishankar et al., 2020; Lim et al., 2024). While NuDFT can also be defined with non-uniform sampling along the time axis, here we focus on non-uniformity along the frequency axis. In SSL, NuDFT enables more effective extraction of IPD features by emphasizing physically informative frequency ranges with densely sampled bins, discussed in Section 2.1.

Let $\mathbf{x}_c \in \mathbb{R}^T$ denote the $T$-sample signal received at the $c$-th channel of a $C$-channel MA, which is a mixture of multiple speakers, noise, and reverberation. The frequency-domain representation $\mathbf{X}_c$ is obtained by applying the LNuDFT independently to each channel, defined as:

$$\mathbf{X}_c\,[k,l] = \sum_{n=0}^{N-1} \underbrace{\mathbf{x}_c\,[n+N_l]\ w\,[n]}_{\text{windowed frame}} \cdot \underbrace{e^{-j2\pi\frac{n}{N}\nu_k}}_{\text{LNuDFT basis}}, \tag{3}$$

where $c$ denotes the channel index, $k \in \{0, \ldots, K-1\}$ the frequency-bin index, and $l$ the frame index. $N$ is the frame length, $N_l$ the starting sample of the $l$-th frame, and $w[\cdot]$ is a window function. $\nu_k$ denotes the location of the $k$-th frequency bin, which maps to the physical frequency $f_k = \frac{\nu_k}{N}f_s$ with sampling frequency $f_s$. When $K = \frac{N}{2} + 1$ and $\nu_k = k$, Eq. (3) reduces to the standard DFT. For the proposed LNuDFT, $\nu_k$ is treated as a learnable parameter instead of being fixed as in the NuDFT, enabling the model to allocate frequency bins more densely in regions that convey physically informative phase cues for SSL. To ensure monotonic ordering and compliance with the Nyquist limit ($0 \le \nu_0 < \ldots < \nu_{K-1} \le \frac{N}{2}$, the maximum representable frequency), $\nu_k$ is defined as the cumulative sum of positive, learnable increments $a_k$ between adjacent bins:

$$\nu_0 = 0, \quad \nu_k = \nu_{k-1} + a_{k-1}, \quad a_k > 0 \ \forall k \in \{1, 2, \ldots, K-1\}. \tag{4}$$

To ensure effective initialization, $\nu_k^{\text{init}}$ and $a_k^{\text{init}}$ are assigned using a logit-based mapping that allocates bins more densely in the mid-frequency range, as follows:

$$\tilde{\nu}_k = \epsilon_{\text{start}} + \frac{\epsilon_{\text{end}} - \epsilon_{\text{start}}}{K-1} k \ \in (\epsilon_{\text{start}}, \epsilon_{\text{end}}), \quad \hat{\nu}_k = \ln\left(\frac{\tilde{\nu}_k}{1 - \tilde{\nu}_k}\right), \tag{5}$$

$$\nu_k^{\text{init}} = \frac{\hat{\nu}_k - \nu_0}{\hat{\nu}_{K-1}} \cdot (K-1), \quad a_{k-1}^{\text{init}} = \nu_k^{\text{init}} - \nu_{k-1}^{\text{init}}, \tag{6}$$

where $0 < \epsilon_{\text{start}} < \epsilon_{\text{end}} < 1$ are hyperparameters that prevent saturation of the logit function. After each gradient update, the following constraints are applied to preserve monotonicity and the Nyquist limit:

$$\hat{a}_k = \min\left(\max\left(\tilde{a}_k, \epsilon_{\min}\right), \epsilon_{\max}\right), \tag{7}$$

$$a_k = \begin{cases} \dfrac{\hat{a}_k}{\sum_{\hat{k}=0}^{K-1} \hat{a}_{\hat{k}}} \cdot \dfrac{N}{2}, & \text{if } \sum_{\hat{k}=0}^{K-1} \hat{a}_{\hat{k}} > \dfrac{N}{2}, \\[2ex] \hat{a}_k, & \text{otherwise}, \end{cases} \tag{8}$$

where $\tilde{a}_k$ is the raw parameter updated by gradient descent, clipped to $(\epsilon_{\min}, \epsilon_{\max})$ to enforce positivity and avoid extreme values. Normalization in Eq. (8) guarantees that $\nu_k$ is bounded by $\frac{N}{2}$. An illustration of LNuDFT is provided in Appendix A.2. Additionally, LNuDFT can be efficiently implemented as 1D convolution (Bagchi & Mitra, 1996a;b), where basis functions serve as convolution kernels of size $N$ and feature dimension $K$, with hop size $H$. This adaptive allocation of frequency bins during training preserves phase information while enhancing robustness and interpretability by focusing on physically informative frequency regions relevant to IPD.

## 3.2 RELATIVE PHASE FEATURE AND MICROPHONE POSITIONAL ENCODING

As MA geometries vary widely across applications, numerous geometry-invariant methods have been explored (Luo et al., 2020; Pandey et al., 2022; Yoshioka et al., 2022; Mu et al., 2024; Xu et al., 2025). In parallel, several geometry-aware approaches have been proposed for SSL (Grinstein et al., 2024; Schwartz et al., 2023; Wang et al., 2024b; Baek et al., 2025). Recognizing that TDOA patterns inherently depend on the MA geometries, GI-DOAEnet (Baek et al., 2025) stands as a representative example, explicitly incorporating geometry information via aMPE and channel-wise multi-head self-attention (CW-MHSA) (Vaswani et al., 2017; Pandey et al., 2022). Here, CW-MHSA computes pairwise channel similarities, while aMPE injects geometry information as positional encodings. This design illustrates how geometry-aware features can be directly embedded into the network to enhance robustness across different MA configurations. Building on this framework, we present AuGeonet, a modified version of GI-DOAEnet that integrates LNuDFT-based GCC-PHAT features and rMPEs, aiming to improve generalization to unseen MA configurations.

**Relative phase features.** To better exploit IPDs, instead of raw DFT coefficients, we adopt GCC-PHAT (Knapp & Carter, 1976) in frequency-domain that emphasizes phase differences while suppressing magnitude variations. Computing all pairwise GCC-PHAT features scales as $\mathcal{O}(C^2)$, so we employ a reference-based scheme that reduces complexity to $\mathcal{O}(C)$. The reference channel $\bar{c}$ is chosen as the microphone closest to the MA center, and the LNuDFT-based GCC-PHAT is defined:

$$\hat{\mathbf{X}}_c^{\text{GCC}}[k,l] = \frac{\mathbf{X}_c[k,l]\,\mathbf{X}_{\bar{c}}^*[k,l]}{|\mathbf{X}_c[k,l]|\,|\mathbf{X}_{\bar{c}}[k,l]|}, \tag{9}$$

where $^*$ denotes complex conjugation and $|\cdot|$ the magnitude. The resulting feature $\hat{\mathbf{X}}^{\text{GCC}} \in \mathbb{C}^{(C-1)\times K \times L}$ is split into real and imaginary parts and concatenated along the frequency axis, producing $\mathbf{X}^{\text{GCC}} \in \mathbb{R}^{(C-1)\times 2K \times L}$. This emphasizes relative phase cues while reducing input dimensionality to $C-1$ by excluding the reference channel.

**Relative microphone positional encoding.** Inspired by the inherently relative nature of IPDs and relative positional encodings (Shaw et al., 2018; Pham et al., 2020; Su et al., 2024), we introduce rMPE, which encodes each microphone's coordinates relative to a reference channel. This design directly aligns with the physics of sound propagation, where TDOA and IPD depend solely on relative microphone positions (see Appendix A.1). Let $\mathbf{p}_c = (x_c, y_c, z_c)$ denote the Cartesian coordinates of microphone $c$. The relative coordinates are:

$$\tilde{x}_c = x_c - x_{\bar{c}}, \quad \tilde{y}_c = y_c - y_{\bar{c}}, \quad \tilde{z}_c = z_c - z_{\bar{c}},$$
$$\tilde{r}_c = \sqrt{\tilde{x}_c^2 + \tilde{y}_c^2 + \tilde{z}_c^2}, \quad \tilde{\vartheta}_c = \operatorname{atan2}(\tilde{y}_c, \tilde{x}_c), \quad \tilde{\varphi}_c = \frac{\pi}{2} - \operatorname{atan2}(\tilde{z}_c, \sqrt{\tilde{x}_c^2 + \tilde{y}_c^2}) \tag{10}$$

where $\tilde{r}_c \in [0,\infty)$, $\tilde{\vartheta}_c \in [-\pi,\pi)$, and $\tilde{\varphi}_c \in [0,\pi]$ represent the distance, azimuth, and elevation, respectively. Following aMPE, rMPE encodes these spherical coordinates into sinusoidal encodings:

$$\mathbf{v}(Q) = \frac{4}{Q}\left[0, 1, \ldots, \frac{Q}{4} - 1\right]^\top \in \mathbb{R}^{\frac{Q}{4}}, \tag{11}$$

$$\mathcal{P}_c^{\text{PM}} = h_{\text{PM}}(\tilde{r}_c, \tilde{\vartheta}_c, \tilde{\varphi}_c; \alpha, \beta, M) \qquad\qquad \mathcal{P}_c^{\text{FM}} = h_{\text{FM}}(\tilde{r}_c, \tilde{\vartheta}_c, \tilde{\varphi}_c; \alpha, \beta, M)$$

$$= \alpha\tilde{r}_c \begin{bmatrix} \cos(2\pi\beta\,\mathbf{v}(M) + \tilde{\vartheta}_c) \\ \sin(2\pi\beta\,\mathbf{v}(M) + \tilde{\vartheta}_c) \\ \cos(2\pi\beta\,\mathbf{v}(M) + \tilde{\varphi}_c) \\ \sin(2\pi\beta\,\mathbf{v}(M) + \tilde{\varphi}_c) \end{bmatrix}, \qquad = \alpha\tilde{r}_c \begin{bmatrix} \cos(\tilde{\vartheta}_c\,\beta\,\mathbf{v}(M)) \\ \sin(\tilde{\vartheta}_c\,\beta\,\mathbf{v}(M)) \\ \cos(\tilde{\varphi}_c\,\beta\,\mathbf{v}(M)) \\ \sin(\tilde{\varphi}_c\,\beta\,\mathbf{v}(M)) \end{bmatrix}, \tag{12}$$

where $\mathbf{v}(Q)$ uniformly partitions the range $[0,1)$ into $\frac{Q}{4}$ elements, and $^\top$ is a transpose operation. $h_{\text{PM}}(\cdot)$ and $h_{\text{FM}}(\cdot)$ denote the phase-modulated (PM) and frequency-modulated (FM) rMPE mapping functions, respectively. $\alpha$ is a scaling factor, $\beta$ a frequency factor, and $M$ the target feature dimension. Stacking across all non-reference microphones yields $\mathcal{P} \in \mathbb{R}^{(C-1)\times M}$, aligned with the input features and CW-MHSA to provide positional cues. With the proposed LNuDFT-based GCC-PHAT and rMPE integrated into AuGeonet, we obtain AGRs, $\mathbf{A}$:

$$\mathbf{A} = \mathcal{A}(\mathbf{x}, \mathbf{p}; \boldsymbol{\Theta}) \in \mathbb{R}^{O \times G \times L}, \tag{13}$$

where $G$ is the AGR dimensionality, $O$ is the number of outputs in the deeply supervised curriculum learning (DSCL) framework (Baek et al., 2023), and $\boldsymbol{\Theta}$ denotes the learnable parameters of

AuGeonet. The architectural details of AuGeonet are provided in Appendix A.3. These AGRs naturally capture audio-geometry relationships and are later integrated into AGG-RL to estimate spatial spectra over candidate DOAs.

## 3.3 AUDIO-GEOMETRY-GRID REPRESENTATION LEARNING

To overcome the limitations of fixed-grid classification, discussed in Section 2.2, we propose AGG-RL, which aligns AGRs with GRs and produces a probabilistic spatial spectrum over arbitrary candidate DOAs by measuring their similarity. We first encode the $d$-th candidate DOA, with azimuth angle $\theta_d$ and elevation angle $\phi_d$, into a $G$-dimensional sinusoidal vector, analogous to Eq. (12) for PM-based rMPE:

$$\hat{\mathbf{G}}_d = h_{\text{Grid}}(\theta_d, \phi_d; \xi, G) = \begin{bmatrix} \cos(2\pi\xi\,\mathbf{v}(G) + \theta_d) \\ \sin(2\pi\xi\,\mathbf{v}(G) + \theta_d) \\ \cos(2\pi\xi\,\mathbf{v}(G) + \phi_d) \\ \sin(2\pi\xi\,\mathbf{v}(G) + \phi_d) \end{bmatrix} \in \mathbb{R}^G, \tag{14}$$

where $h_{\text{Grid}}(\cdot)$ denotes the grid encoding function, and $\xi$ is a modulation frequency. Using Grid-net $\mathcal{G}_o(\cdot)$, a simple DNN with learnable parameters $\mathbf{\Psi}_o$ specific to each output index $o$ (see Appendix A.4), $\hat{\mathbf{G}}_d$ is transformed into a GR:

$$\mathbf{G}_{d,o} = \mathcal{G}_o\Big(\hat{\mathbf{G}}_d; \mathbf{\Psi}_o\Big) \in \mathbb{R}^G. \tag{15}$$

Given AGRs $\mathbf{A} \in \mathbb{R}^{O \times G \times L}$ from AuGeonet, the similarity is computed by a scaled inner product followed by a sigmoid function $\sigma(\cdot)$, producing the probabilistic spatial spectrum:

$$\hat{\mathbf{S}}_{d,o,l} = \sigma\left(\frac{\mathbf{G}_{d,o}^\top \mathbf{A}_{o,l}}{\sqrt{G}}\right) \in [0, 1], \tag{16}$$

where scaling by $\sqrt{G}$ stabilizes optimization by controlling the variance of the inner product (Vaswani et al., 2017). Through this procedure, AGRs are encouraged to align with GRs at true-source DOAs while diverging from non-source directions. Meanwhile, GRs are trained to represent candidate DOAs independently of the audios and MA geometries, enabling flexible-grid SSL.

Candidate DOAs are sampled using Fibonacci sphere points (Saff & Kuijlaars, 1997), providing near-uniform coverage over the unit sphere. During training, Fibonacci grids with $D$ points are randomly rotated for data augmentation. Soft targets are defined as oracle spatial spectra with varying beamwidths (Baek et al., 2023) and used in a weighted binary cross-entropy (BCE) loss (Nguyen et al., 2020), which emphasizes positive samples. At inference, an iterative peak detection algorithm (Baek et al., 2023) is applied to the final layer output to identify multiple source DOAs. Further details are given in Appendix A.5–A.8. In summary, AGG-RL enables flexible-grid and geometry-invariant SSL without retraining, while retaining the interpretability of classification-based methods.

## 4 EXPERIMENTAL SETUP

### 4.1 MODEL IMPLEMENTATION AND BASELINES

We compared our method against classical baselines (MUSIC (Schmidt, 1986), SRP-PHAT (Dibiase, 2000)) and recent DNN-based methods (Unet (Schwartz et al., 2023), Neural-SRP (Grinstein et al., 2024), GI-DOAEnet (Baek et al., 2025)). To ensure fair comparison, all methods were implemented in a causal setting with identical DFT parameters ($N = 512$, $K = 257$, $H = 128$) and a Hann window. For classical methods, candidate DOAs were sampled using Fibonacci grids with $D = 512$ and 2048. Unet and Neural-SRP were modified following GI-DOAEnet to produce multiple spatial spectra outputs, following Baek et al. (2025). Unet was re-implemented from the original paper, while Neural-SRP[1] and GI-DOAEnet[2] were adapted from public code. Unet used SRP-PHAT input with $D = 512$, whereas Neural-SRP employed time-domain GCC-PHAT features; both were

---

[1]https://github.com/egrinstein/neural_srp
[2]https://github.com/BaekMS/GI-DOAEnet

adapted to output spectra on a Fibonacci grid with $D = 2048$. For GI-DOAEnet, both PM- and FM-based aMPE variants were evaluated, with an output dimension of $D = 2048$. We additionally note that IPDnet with template matching (Wang et al., 2024b) was excluded from direct comparison due to its impractical computational cost (23.2 GFLOPs for 2-channel input, making it prohibitive for larger MAs) and incompatible output format (estimates microphone pairwise, frequency-dependent IPDs instead of DOAs). For the proposed method, FM-based rMPE was adopted as default, as it slightly outperformed PM-based encoding in preliminary experiments, and $D = 2048$ was used for consistency. For LNuDFT of AuGeonet, initialization was performed with $\epsilon_{\text{start}} = 0.15$ and $\epsilon_{\text{end}} = 0.95$, and constraints were applied with $\epsilon_{\text{min}} = 0.01$ and $\epsilon_{\text{max}} = 100$. All DNN-based methods followed the same training setup described in Appendix A.9.

## 4.2 DATASET AND EVALUATION METRICS

Table 1: Parameters used for synthetic dataset generation.

| Parameter | Interval | Unit |
|---|---|---|
| Number of speakers | [1, 2] | - |
| RT60 | [0.2, 1.3] | s |
| Room size | [3×3×2.5, 10×8×6] | $m^3$ |
| Distance | [0.3, 2.5] | m |
| Azimuth | [0, 360) | ° |
| Elevation | [0, 180] | ° |
| SNR (speech vs. noise) | [-5, 30] | dB |
| SIR (between speakers) | [-5, 5] | dB |
| SIR (between noises) | [-15, 15] | dB |

Synthetic datasets were used for training, validation, and partial evaluation. The training set was generated on-the-fly to provide diverse conditions, with a fixed number of channels per batch. Table 1 summarizes the ranges of acoustic and spatial parameters used to generate the synthetic mixtures, including reverberation time (RT60), signal-to-noise ratio (SNR) between speech and noise, and signal-to-interference ratio (SIR) between speakers or between noise sources. For each sample, we simulated multichannel room impulse responses (RIRs) using the image source method (Allen & Berkley, 1979) implemented in gpuRIR (Diaz-Guerra et al., 2021b). Speech signals were drawn from LibriSpeech (Panayotov et al., 2015) and noise signals from MS-SNSD (Reddy et al., 2019), each cropped to 4 seconds. Additional implementation details for the synthetic data generation are provided in Appendix A.10.

Table 2: Evaluation dataset configuration.

| Dataset | Real Dataset | Microphone Array | Channels | Training Exposure |
|---|---|---|---|---|
| *NAO robot* | ✓ | NAO robot | 12 | ✓ (seen) |
| *Eigenmike* | ✓ | Eigenmike | 32 | ✗ (unseen) |
| *Dynamic-S* | ✗ | Dynamic | 4–12 | ✓ (seen) |
| *Dynamic-U* | ✗ | Dynamic | 13–16 | ✗ (unseen) |

After training, the model with the lowest validation loss was selected for evaluation. Evaluation was conducted using four datasets, as shown in Table 2. Real recordings were taken from LO-CATA (Löllmann et al., 2018), where *NAO robot* and *Eigenmike* datasets consisted of recordings using the NAO robot[3] and the Eigenmike[4], respectively. Only non-moving sources with up to two speakers were used for evaluation, resampled to 16 kHz. Two synthetic datasets, *Dynamic-S* and *Dynamic-U*, were constructed to evaluate generalization to seen and unseen numbers of channels, respectively, across a variety of MA geometries. These synthetic datasets were generated with 300 utterances per channel, following the same procedure as the training set but using different source material: TIMIT (Garofolo et al., 1993) for speech and ESC-50 (Piczak, 2015) for noise. For evaluation metrics, we used mean absolute error (MAE, °) and accuracy ($\text{ACC}_{10}$, %) (see Appendix A.11). MAE measured the average angular error; lower values indicate better performance. $\text{ACC}_{10}$ measured the percentage of predictions within a $10°$ margin; higher values indicate better performance. Both metrics were computed per utterance and averaged over each set.

---

[3] https://www.aldebaran.com/en/nao
[4] https://eigenmike.com/eigenmike-em32

## 5 RESULTS AND DISCUSSION

Table 3: Experimental results with different methods. The best results on each dataset are in **bold**.

| | NAO robot | | Eigenmike | | Dynamic-S | | Dynamic-U | |
|---|---|---|---|---|---|---|---|---|
| | MAE | ACC$_{10}$ | MAE | ACC$_{10}$ | MAE | ACC$_{10}$ | MAE | ACC$_{10}$ |
| MUSIC$_{512}$ | 20.63 ±2.44 | 64.95 | 29.93 ±3.09 | 36.37 | 30.35 ±0.63 | 27.94 | 27.13 ±0.89 | 33.20 |
| MUSIC$_{2048}$ | 19.31 ±2.65 | 69.77 | 29.78 ±3.06 | 36.68 | 30.24 ±0.63 | 28.50 | 26.88 ±0.90 | 33.92 |
| SRP-PHAT$_{512}$ | 22.36 ±1.79 | 67.95 | 27.45 ±1.73 | 41.38 | 43.98 ±0.70 | 24.55 | 38.64 ±1.03 | 32.13 |
| SRP-PHAT$_{2048}$ | 21.77 ±1.73 | 67.84 | 26.88 ±1.93 | 53.22 | 43.89 ±0.70 | 25.10 | 38.40 ±1.04 | 32.39 |
| Unet | 10.89 ±1.53 | 86.25 | 14.89 ±1.76 | 65.82 | 19.94 ±0.69 | 58.88 | 19.15 ±0.94 | 60.57 |
|   with AGG-RL | 12.79 ±3.26 | 77.33 | 16.86 ±3.26 | 55.69 | 21.23 ±0.69 | 53.74 | 20.11 ±0.95 | 56.08 |
| Neural-SRP | 9.72 ±2.28 | 78.66 | 52.75 ±18.61 | 22.16 | 19.60 ±0.74 | 52.32 | 21.18 ±1.01 | 45.51 |
|   with AGG-RL | 9.89 ±2.55 | 72.80 | 13.25 ±2.32 | 37.07 | 19.79 ±0.74 | 50.56 | 19.05 ±1.01 | 54.13 |
| GI-DOAEnet$^{FM}$ | 11.31 ±2.54 | 77.36 | 93.61 ±13.06 | 0.00 | 15.49 ±0.55 | 64.36 | 54.81 ±1.73 | 6.10 |
| GI-DOAEnet$^{PM}$ | 12.26 ±2.27 | 72.47 | 77.09 ±13.32 | 2.82 | 17.40 ±0.59 | 58.54 | 79.17 ±1.81 | 0.58 |
| Proposed | **8.25** ±1.52 | **90.78** | **11.24** ±1.76 | **72.17** | 10.32 ±0.49 | 77.34 | 14.12 ±0.77 | 63.17 |
|   (i) rMPE-PM | 8.38 ±1.63 | 89.85 | 13.42 ±2.00 | 70.09 | 11.55 ±0.50 | 74.46 | 12.46 ±0.78 | 67.97 |
|   (ii) DFT | 13.24 ±2.97 | 72.32 | 111.21 ±9.86 | 0.00 | 16.47 ±0.58 | 60.83 | 87.71 ±1.89 | 0.92 |
|   (iii) DFT + GCC-PHAT | 9.17 ±3.44 | 87.74 | 16.53 ±3.97 | 39.35 | 10.26 ±0.48 | 77.56 | 17.90 ±0.71 | 45.11 |
|   (iv) LNuDFT + Uniform init. | 8.69 ±2.97 | 90.03 | 15.13 ±2.68 | 40.70 | 10.44 ±0.47 | 77.69 | 23.03 ±0.89 | 36.05 |
|   (v) NuDFT + Logit init. | 8.96 ±2.83 | 89.34 | 17.34 ±2.47 | 28.52 | 10.64 ±0.48 | 76.56 | **11.83** ±0.66 | **72.77** |
|   (vi) Fixed grid | 9.17 ±2.28 | 87.31 | 13.58 ±2.60 | 40.82 | **9.57** ±0.46 | **81.70** | 13.84 ±1.07 | 65.29 |
|   (vii) Gridnet with FM Enc. | 9.95 ±2.36 | 85.60 | 12.58 ±2.68 | 50.55 | 11.10 ±0.49 | 75.49 | 15.24 ±0.68 | 58.60 |
|   (viii) Gridnet with Cartesian | 9.08 ±2.85 | 88.17 | 11.87 ±2.80 | 64.62 | 10.41 ±0.48 | 77.28 | 23.10 ±0.91 | 34.26 |

Table 3 presents the overall results on the evaluation datasets. Rows above the double line correspond to baseline methods, while rows below represent the proposed method and its ablation studies. The 95% confidence intervals (CIs) for the MAE are reported using ± values, which indicate the corresponding margin of error.

**Comparison with baselines.** The subscripts 512 and 2048 indicate the number of grids used for SRP-PHAT and MUSIC. Both methods consistently underperformed compared to DNN-based approaches across all datasets. Moreover, increasing the number of grid points did not lead to meaningful improvements. Accordingly, Unet adopted SRP-PHAT with $D = 512$ for efficiency. Among the DNN-based baselines, Neural-SRP achieved lower MAE than Unet on the *NAO robot* and *Dynamic-S* datasets, while Unet outperformed Neural-SRP on the remaining datasets and metrics. Notably, Neural-SRP exhibited substantial performance degradation on *Eigenmike*. When AGG-RL was applied, performance of Unet slightly degraded overall. In contrast, Neural-SRP showed performance drops in seen conditions such as *NAO robot* and *Dynamic-S*, but significant improvements under unseen conditions, including *Eigenmike* and *Dynamic-U*. These results demonstrate that AGG-RL effectively enhanced generalizability for Neural-SRP. GI-DOAEnet variants with different aMPE types outperformed Unet and Neural-SRP on *Dynamic-S*, which shared the same channel configuration as the training set. However, their performance dropped significantly under unseen settings, including *Eigenmike* and *Dynamic-U*.

The proposed method consistently achieved superior performance across all datasets, clearly validating its robustness and effectiveness. However, it achieved slightly worse results on unseen conditions compared to the seen conditions. This indicates a performance gap between seen and unseen scenarios but still outperformed all baselines. To further validate these findings, we provide box and violin plots (Appendix A.12) to examine the distribution of errors, results under different acoustic conditions (Appendix A.13) to assess robustness across diverse environments, and spatial spectrum visualizations (Appendix A.15) to illustrate the interpretability and stability of the proposed method. Additionally, Appendix A.14 reports results on the STARSS23 dataset (Shimada et al., 2023), further demonstrating the effectiveness of the method on a real-world benchmark with different recording conditions. The following ablation studies further clarify the contributions of each component.

**Effectiveness of GCC-PHAT and rMPE.** Experiment (i) replaced rMPE with PM-based one. This variant showed comparable but slightly inferior performance on most datasets, except for *Dynamic-U*, indicating that FM was more effective than PM for rMPE. Accordingly, all proposed and subsequent ablation studies adopted FM-based rMPE as the default. In experiment (ii), both GCC-PHAT and rMPE were replaced with the standard DFT and FM-based aMPE, respectively. This led to a consistent performance drop across all datasets, underscoring the importance of relative representations introduced by GCC-PHAT and rMPE. In both GI-DOAEnet and AuGeonet,

CW-MHSA layers were used to capture channel-wise relationships. However, prior works (Kazemnejad et al., 2023; Zhou et al., 2024) have shown that MHSA performance degrades when extrapolating to longer sequences than those encountered during training. We therefore hypothesize that the relative nature of GCC-PHAT and rMPE mitigated this limitation, thereby enhancing generalization to unseen conditions.

**Effectiveness of LNuDFT.** Experiments (iii)–(v) evaluated the contribution of LNuDFT under different configurations. In experiment (iii), LNuDFT was replaced with the standard DFT while retaining GCC-PHAT and rMPE. Performance degraded on all datasets except *Dynamic-S*, validating the effectiveness of LNuDFT. Experiment (iv) initialized LNuDFT with the uniform spacing, identical to the standard DFT and allowed it to be trained. While it achieved comparable results to the proposed method on *Dynamic-S* and *NAO robot*, performance decreased on *Eigenmike* and *Dynamic-U*. In experiment (v), NuDFT parameters were initialized using a proposed logit-based mapping and then frozen during training. This variant yielded the best performance on *Dynamic-U*, suggesting that initializing LNuDFT with an informative frequency distribution benefited generalization to unseen conditions. While logit-based initialization was more effective in practice, the choice of mapping function and hyperparameters was made empirically. Identifying optimal initialization strategies remains an open direction for future research and may further enhance performance. The visualization of trained LNuDFT parameters (Appendix A.2) reveals that they densely focused on physically informative frequency regions, enhancing both model interpretability and robustness.

**Effectiveness of AGG-RL.** In experiment (vi), AGG-RL was replaced with a fixed-grid setup ($D = 2048$), directly predicting spatial spectra as in standard classification models. This variant achieved the best performance on *Dynamic-S*, which matched the training condition, and slightly outperformed the proposed method on *Dynamic-U*. However, it degraded on real datasets, showing that AGG-RL was crucial for generalization. Experiments (vii) and (viii) replaced Gridnet inputs with FM-based encodings of Eq. (12) and raw Cartesian coordinates, respectively. FM-based encoding achieved similar results, but PM-based encoding was slightly superior overall. Raw Cartesian inputs yielded comparable performance on most datasets but dropped on *Dynamic-U*, suggesting that sinusoidal encodings better captured the candidate DOA grid structure.

Table 4: Experimental results with different numbers of candidate grids $D$. The best results on each dataset are in **bold**.

| $D$ | NAO robot | | Eigenmike | | Dynamic-S | | Dynamic-U | |
| --- | --- | --- | --- | --- | --- | --- | --- | --- |
| | MAE | $\overline{ACC}_{10}$ | MAE | $\overline{ACC}_{10}$ | MAE | $\overline{ACC}_{10}$ | MAE | $\overline{ACC}_{10}$ |
| 128 | 12.69 $\pm 3.14$ | 68.01 | 13.60 $\pm 2.83$ | 58.65 | 13.84 $\pm 0.45$ | 60.29 | 16.34 $\pm 0.72$ | 45.70 |
| 256 | 10.92 $\pm 2.79$ | 81.25 | 13.51 $\pm 3.15$ | 58.20 | 13.11 $\pm 0.51$ | 69.79 | 16.32 $\pm 0.80$ | 51.33 |
| 512 | 10.19 $\pm 2.17$ | 87.51 | 11.65 $\pm 2.72$ | 61.06 | 11.59 $\pm 0.49$ | 74.60 | 14.97 $\pm 0.77$ | 61.27 |
| 1024 | 9.21 $\pm 2.65$ | 89.91 | 12.13 $\pm 2.86$ | 72.66 | 10.66 $\pm 0.49$ | 77.52 | 14.24 $\pm 0.76$ | 61.88 |
| 2048 | **8.25** $\pm 1.52$ | **90.78** | **11.24** $\pm 1.76$ | 72.17 | 10.32 $\pm 0.49$ | 77.34 | 14.12 $\pm 0.77$ | 63.17 |
| 4096 | 9.16 $\pm 2.56$ | 89.91 | 11.51 $\pm 2.68$ | 73.06 | 10.14 $\pm 0.50$ | 77.73 | 14.01 $\pm 0.77$ | 63.72 |
| 8192 | 8.91 $\pm 2.63$ | 89.54 | 11.52 $\pm 2.76$ | **75.97** | **9.91** $\pm 0.49$ | 78.12 | **13.85** $\pm 0.76$ | 63.83 |
| 16384 | 8.79 $\pm 2.63$ | 89.04 | 11.54 $\pm 2.80$ | 72.48 | 9.96 $\pm 0.50$ | **78.21** | 13.87 $\pm 0.77$ | **64.18** |

Table 4 further reports the results of the proposed method with varying $D$. When $D$ was too small, performance dropped due to limited spatial resolution. Once $D \geq 512$, performance stabilized across all datasets, confirming the grid-flexibility of AGG-RL. When $D$ exceeded 2048, the trends between the real and synthetic datasets diverged slightly. For the synthetic datasets (*Dynamic-S* and *Dynamic-U*), the performance continued to improve mildly with larger $D$. Since these datasets were closely matched to the simulated training domain, finer angular discretization reduced quantization error and provided consistent, though marginal, improvements. This indicates diminishing returns beyond a certain resolution. In contrast, performance on the real datasets (*NAO robot* and *Eigenmike*) showed a slight degradation once $D$ exceeded 2048 (except for $ACC_{10}$ on *Eigenmike*). Real-world recordings inevitably contained small acoustic mismatches—such as sensor noise, microphone manufacturing tolerances, and RIR deviations—that were not present in synthetic data. With an overly dense grid, the localization decision became more sensitive to these perturbations, increasing estimation variance. Overall, the results suggest that increasing $D$ beyond a certain threshold yields limited gains, and excessively large $D$ is not necessarily beneficial in practice. t-SNE visualizations of the GRs (Appendix A.16) further confirm that Gridnet learned structured spatial embeddings. Overall, AGG-RL enhanced robustness on real recordings and unseen conditions, effectively integrating audio-geometry-grid information into a shared latent space.

Table 5: Number of parameters and FLOPs under different configurations.

(a) Models with varying numbers of channels

| Model | Params (M) | FLOPs (G) | | |
|---|---|---|---|---|
| | | $C = 4$ | $C = 8$ | $C = 12$ |
| Unet | 8.40 | 1.14 | 1.49 | 2.10 |
| Neural-SRP | 1.13 | 1.86 | 8.54 | 20.08 |
| GI-DOAEnet | 2.32 | 0.49 | 0.89 | 1.29 |
| AuGeonet | | | | |
|    with Fixed Grid | 2.32 | 0.42 | 0.82 | 1.22 |
|    with AGG-RL | 1.86 | 0.43 | 0.83 | 1.23 |

(b) Gridnet with varying grid sizes

| $D$ | Gridnet Params (M) | GR Params (M) | FLOPs (G) |
|---|---|---|---|
| 1024 | 1.19 | 0.26 | 0.27 |
| 2048 | 1.19 | 0.52 | 0.54 |
| 4096 | 1.19 | 1.04 | 1.07 |

**Computational resource comparison.** We measured inference complexity using fvcore[5], with 1-second inputs, defaulting to $G = 256$ and $D = 2048$. Table 5(a) compares parameters and FLOPs across different numbers of channels, while Table 5(b) reports Gridnet results with varying grid sizes. Unet had the largest parameter count and Neural-SRP the highest FLOPs, with FLOPs of both models being scaled quadratically with $C$ due to pairwise features. GI-DOAEnet and AuGeonet with fixed grid shared the same parameters, but AuGeonet achieved lower FLOPs with reference-based GCC-PHAT, also ensuring linear scaling in $C$. With AGG-RL, AuGeonet further reduced parameters by replacing direct $D = 2048$ mapping with a $G = 256$ representation; FLOPs increased only slightly from the similarity computation in Eq. (16). For Gridnet with GRs, FLOPs scaled linearly with $D$, and both FLOPs and parameter counts were on the same order as AuGeonet, contributing to the overall computational complexity. Nevertheless, FLOPs remained lower than those of baselines and IPDnet (Wang et al., 2024b). In practice, the overhead from Gridnet is manageable: GRs can be cached for static environments, or $D$ can be reduced with minimal performance loss. Further studies on efficiency improvements, including lightweight design alternatives, are left for future work.

## 6 CONCLUSION

In this paper, we proposed AGG-RL, a novel framework for grid-flexible and geometry-invariant SSL that jointly learns complementary representations from audio-geometry-grid inputs. To improve generalization and interpretability, we introduced two physics-informed components: LNuDFT, which enables adaptive frequency analysis to enhance IPD distinguishability, and rMPE, which encodes relative microphone position cues. Extensive experiments across multiple datasets demonstrated that our framework achieved superior generalizability and robustness, particularly under unseen geometries and real-world recordings. We believe that AGG-RL provides a solid foundation for advancing 3D acoustic scene understanding in practical environments. For future work, we plan to extend AGG-RL to multimodal scenarios and broader spatial sensing tasks that require scalable grid or geometry configurations. We also aim to theoretically analyze the generalization properties of LNuDFT-based representations, providing a stronger foundation for physics-informed DNNs.

REPRODUCIBILITY STATEMENT

In this paper, we rely on publicly available datasets and open-source toolkits to ensure reproducibility. Detailed descriptions of the datasets, libraries, and implementation details are provided in the main text and the Appendix. The source code is publicly available at `https://github.com/BaekMS/Audio-Geometry-Grid_Representation-Learning`.

ACKNOWLEDGMENTS

This work was partly supported by the National Research Foundation of Korea(NRF) grant funded by the Korea government(MSIT) (RS-2025-00557944) and Institute of Information & communications Technology Planning & Evaluation (IITP) under the artificial intelligence semiconductor support program to nurture the best talents (IITP-2026-RS-2023-00253914) grant funded by the Korea government(MSIT).

---

[5]`https://github.com/facebookresearch/fvcore`

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

# A  APPENDIX

## A.1  RELATIONSHIP BETWEEN TDOA AND RELATIVE MICROPHONE POSITIONS

We derive here the relationship between the TDOA and relative microphone positions under the standard plane-wave assumption. Assuming a far-field source, the time of arrival $\tau(\cdot)$ at a microphone position $\mathbf{p} \in \mathbb{R}^3$ is given by

$$\tau(\mathbf{p}) = \tau_0 - \frac{\mathbf{u}^\top \mathbf{p}}{v}, \tag{17}$$

where $\tau_0$ is the reference arrival time at the origin of the coordinates, $\mathbf{u} \in \mathbb{R}^3$ is the unit direction vector of the incoming wave, and $v$ is the speed of sound. Let $\mathbf{p}_m$ and $\mathbf{p}_n$ denote the positions of the $m$-th and $n$-th microphones, respectively. Their TDOA $\Delta\tau_{m,n}$ is

$$\Delta\tau_{m,n} = \tau(\mathbf{p}_m) - \tau(\mathbf{p}_n) = \frac{\mathbf{u}^\top(\mathbf{p}_n - \mathbf{p}_m)}{v} = \frac{\mathbf{u}^\top \Delta\mathbf{p}_{m,n}}{v}, \tag{18}$$

where $\Delta\mathbf{p}_{m,n} := \mathbf{p}_n - \mathbf{p}_m$ is the relative microphone position vector. This result shows that the TDOA depends solely on the relative microphone positions $\Delta\mathbf{p}_{m,n}$, which is analogous to Eq. (10), not on their absolute coordinates. As the IPD is given by $\Delta\theta_{m,n}(f) = 2\pi f \Delta\tau_{m,n}$, the same property holds for IPDs as well. Therefore, encoding microphone geometry in a relative form, as done in the proposed rMPE, is directly grounded in the underlying physics of wave propagation.

## A.2  LNuDFT PARAMETER VISUALIZATION

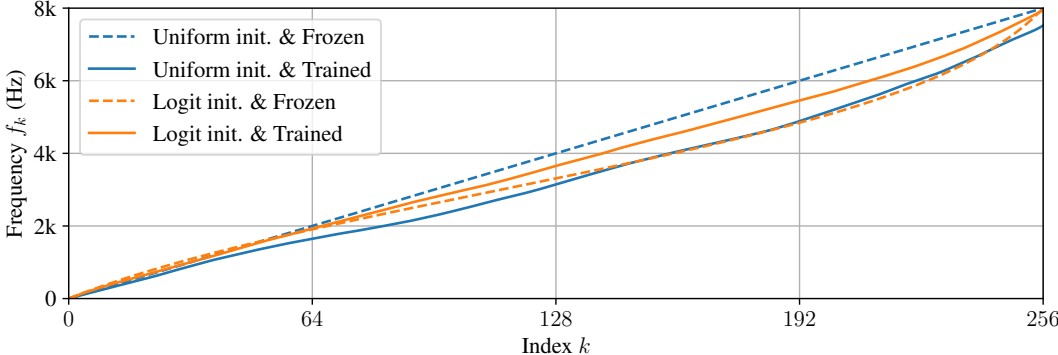

Figure 3: Frequency response of LNuDFT parameters.

Figure 3 shows the frequency response of the LNuDFT parameters with $f_k = \frac{\nu_k}{N} f_s$, where the sampling frequency was $f_s = 16$ kHz. The horizontal axis represents the indices of the LNuDFT parameters, $k$, and the vertical axis represents the frequency $f_k$ in Hz. The standard DFT was initialized with uniformly spaced bins, which after training evolved into a non-uniform spacing that allocates higher resolution to the 1.5–7.5 kHz range. In contrast, the proposed logit-based initialization already placed parameters densely within the 2–6 kHz region at initialization, and they tended to evolve toward more uniform spacing after training. This behavior indicates that the model tended to attenuate resolution at too low and high frequencies, concentrating more bins in the mid-frequency range. It denotes that the proposed initialization and the results of LNuDFT were well aligned with the spectral characteristics of speech signals, which are primarily concentrated in this frequency range, and provides informative IPD cues crucial for SSL and TDOA estimation.

The selected values of $\epsilon_{\text{start}} = 0.15$ and $\epsilon_{\text{end}} = 0.95$ for the logit-based initialization were chosen to assign denser sampling to the low- and mid-frequency regions, while sparsely covering the high-frequency region to avoid spatial aliasing. As shown in Fig. 3, this initialization results in frequency allocations that concentrate primarily on the mid-frequency bins where IPD cues are most reliable. Furthermore, the initialization also assigned slightly denser sampling in the low-frequency bins than in the high-frequency bins, ensuring stable IPD information while avoiding aliasing effects. We performed a grid search over $\epsilon_{\text{start}}$ and $\epsilon_{\text{end}}$ and observed that small perturbations around the selected values did not significantly affect performance. The chosen configuration provided the most consistent and robust results across our experiments.

## A.3 AuGeonet Architecture

Figure 4: AuGeonet architecture.

The overall structure of AuGeonet is modified from GI-DOAEnet (Baek et al., 2025), with detailed architecture available in the original paper. The default settings of the AuGeonet were $M = 128$, $\alpha = 7$, $\beta = 4$, $O = 3$, and $G = 256$. Other hyperparameters followed those of GI-DOAEnet. Figure 4 illustrates the modified AuGeonet architecture, and the key modifications are summarized below:

- **Input feature.** The input is replaced with LNuDFT-based GCC-PHAT features.

- **Integrating the geometry information.** Each channel of the input GCC-PHAT features is treated as a batch. These features are first processed with 1D batch normalization (BN) (Ioffe & Szegedy, 2015), and relative microphone coordinates in both Cartesian and sinusoidal spherical forms $(\tilde{x}_c, \tilde{y}_c, \tilde{z}_c, \tilde{r}_c, \sin \tilde{\vartheta}_c, \cos \tilde{\vartheta}_c, \sin \tilde{\varphi}_c, \cos \tilde{\varphi}_c)$ from Eq. (10) are concatenated along the frequency dimension to provide comprehensive geometry information. Then, an initial ConvBlock (CB), consisting of a 1D convolution layer, an ELU (Clevert et al., 2016) activation, and a BN, projects the frequency dimension from $2K+8$ to feature size $M$. A stack of Residual ConvBlocks (RCBs), each composed of two CBs with a residual connection, follows the initial CB to extract local features. rMPE is added to the input of each RCB, and a total of 4 RCBs are used.

  After the RCBs, four spatio-temporal dual-path blocks (STDPBs) are applied. Each STDPB consists of a CW-MHSA with rMPE, which captures spatial dependencies by applying MHSA across the channel dimension for each time frame, and a frame-wise GRU (Cho et al., 2014), which models temporal dependencies.

- **Output representation.** Finally, $O$ representation mapping blocks (RMBs) are used instead of the spatial spectrum mapping blocks in GI-DOAEnet. Each RMB consists of an RCB, which applies layer normalization (LN) (Ba et al., 2016) instead of BN, followed by a linear layer that projects the features to size $G$. The output of RCB each RMB is also passed to the subsequent one.

## A.4 GRIDNET ARCHITECTURE

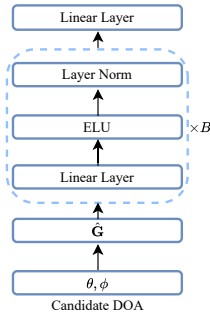

Figure 5: Gridnet architecture.

Gridnet consists of $B$ sequential blocks, each composed of a linear layer that preserves the feature dimension $G$, followed by an ELU activation (Clevert et al., 2016) and an LN layer, as illustrated in Fig. 5. A final linear layer produces the output, GR. All candidate DOAs are modulated with sinusoidal encoding before being input to Gridnet. These modulated candidate DOAs are processed in batch, allowing Gridnet to efficiently handle varying numbers of candidates $D$ without retraining. $B$ was set to 3, and the frequency modulation factor $\xi$ was set to 1 in our experiments.

## A.5 FIBONACCI SPHERE SAMPLING

---

**Algorithm 1:** Fibonacci sphere point generation

**Input:** Number of DOA grids $D$

**Output:** Set of 3D points $\mathbf{\Psi} = \{\psi_d\}_{d=0}^{D-1}$, $\psi_d = (x_d, y_d, z_d, \theta_d, \phi_d)$ on the unit sphere

$\varphi \leftarrow \frac{1+\sqrt{5}}{2}$

$\vartheta \leftarrow \frac{2\pi}{\varphi}$

**for** $d \leftarrow 0$ **to** $D - 1$ **do**

$\quad z \leftarrow 1 - \frac{2d+1}{D}$

$\quad r \leftarrow \sqrt{1 - z^2}$

$\quad \omega \leftarrow d \cdot \vartheta$

$\quad x \leftarrow r \cos \omega$

$\quad y \leftarrow r \sin \omega$

$\quad \theta \leftarrow \text{atan2}(y, x)$

$\quad \phi \leftarrow \frac{\pi}{2} - \text{atan2}(z, \sqrt{x^2 + y^2})$

$\quad$ Add point $\psi_d = (x, y, z, \theta, \phi)$ to $\mathbf{\Psi}$

**end**

**return** $\mathbf{\Psi}$

---

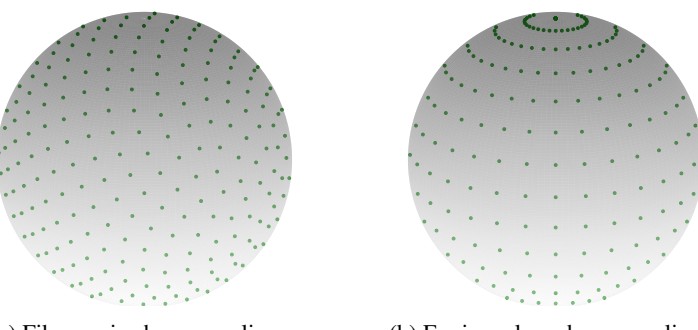

(a) Fibonacci sphere sampling.      (b) Equiangular sphere sampling.

Figure 6: Examples of (a) Fibonacci sphere sampling and (b) equiangular sphere sampling.

Fibonacci sphere sampling generates nearly uniform points on the unit sphere using the golden angle, thereby avoiding the pole clustering problem of latitude–longitude grids (Saff & Kuijlaars, 1997). Algorithm 1 summarizes the procedure, with $\varphi$ denoting the golden ratio. Figure 6(a) shows an example with $D = 512$ points, while Fig. 6(b) illustrates equiangular sampling (32 azimuth × 16 elevation), which suffers from dense clustering at the poles. This method efficiently approximates uniform sampling without complex computations.

## A.6 ORACLE SPATIAL SPECTRUM GENERATION

The oracle spatial spectrum $\mathbf{S} \in \mathbb{R}^{D \times O \times L}$ is generated as:

$$\mathbf{S}_{d,o,l} = \begin{cases} \max\limits_{\psi_l \in \Psi_l} \left\{ e^{\kappa(\gamma_o)\left(\cos\left(\delta(\varsigma_d, \psi_l)\right) - 1\right)} \right\}, & \text{if } |\Psi_l| > 0, \\ 0, & \text{otherwise,} \end{cases} \tag{19}$$

$$\delta(\varsigma, \psi) = \arccos\left(x_\varsigma x_\psi + y_\varsigma y_\psi + z_\varsigma z_\psi\right), \tag{20}$$

$$\kappa(\gamma) = \frac{-\ln\sqrt{2}}{\cos\gamma - 1}, \tag{21}$$

where $\Psi_l$ is the set of ground-truth DOAs at the $l$-th frame, and $\varsigma_d$ is the $d$-th DOA candidate. The angular distance between a candidate and a ground-truth DOA is computed by $\delta(\cdot)$, and $\gamma_o$ controls the beamwidth (Choi & Chang, 2022) for the $o$-th output. A smaller $\gamma_o$ yields narrower beams with sharper peaks but weaker relationships across neighboring DOAs, while a larger $\gamma_o$ produces wider beams with smoother transitions.

## A.7 LOSS FUNCTION

The weighted BCE loss is defined as:

$$\mathcal{L}(\mathbf{S}, \hat{\mathbf{S}}) = -\frac{1}{L \cdot D} \sum_{o=1}^{O} \sum_{l=1}^{L} \sum_{d=1}^{D} \left\{ \rho \cdot \mathbf{S}_{d,o,l} \log \hat{\mathbf{S}}_{d,o,l} + (1 - \mathbf{S}_{d,o,l}) \log\left(1 - \hat{\mathbf{S}}_{d,o,l}\right) \right\}, \tag{22}$$

where $\rho$ is a weighting factor to balance the loss between positive (ground-truth DOAs) and negative samples (Nguyen et al., 2020). In our experiments, we set $\rho = 2$.

## A.8 ITERATIVE MAX-PEAK SELECTION ALGORITHM

---
**Algorithm 2:** Iterative max-peak selection from spatial spectrum

---
**Input:** Predicted spatial spectrum $\hat{\mathbf{S}}_{o,l}$, number of active speakers $T_l$, angular distance margin $\bar{L}$
**Output:** Set of estimated DOAs $\hat{\Psi}_l$
Initialize $\hat{\Psi}_l \leftarrow \emptyset$
**while** $|\hat{\Psi}_l| < T_l$ **do**
    Find index $d^* \leftarrow \arg\max_d \hat{\mathbf{S}}_{d,o,l}$
    Add DOA candidate $\psi_{d^*}$ to $\hat{\Psi}_l$
    Suppress $\hat{\mathbf{S}}_{d,o,l} \leftarrow 0$ for all $d$ within angular distance $\bar{L}$ of $d^*$ (computed using Eq. (20)).
**return** $\hat{\Psi}_l$

---

Algorithm 2 outlines the iterative max-peak selection method for estimating DOAs (Baek et al., 2023) from the predicted spatial spectrum $\hat{\mathbf{S}}_{o,l} \in \mathbb{R}^D$, where $D$ is the number of candidate points at the $l$-th frame and $o$-th output. The number of active speakers $T_l$ is assumed to be known a priori, and $\bar{L}$ is an angular distance margin to avoid selecting multiple peaks from the same source, which was set to $10°$ in our experiments.

## A.9 MODEL TRAINING DETAILS

Every DNN-based method used similar training settings. The batch size was set to 16 for most models, except Neural-SRP which reduced to 1 due to memory constraints of GPU. All models

were implemented in PyTorch and trained on a single NVIDIA RTX 3090 / 4090 GPU. We used the weighted BCE loss in Eq. (22), and the Adam optimizer (Kingma & Ba, 2014), gradient clipping within 1, and an adaptive learning rate schedule that decayed by 0.9 if the validation loss did not improve for two consecutive epochs. To facilitate robust SSL learning, we employed complexity gradual training (CGT) strategy (Baek et al., 2025), consisting of multi-stage geometry learning (MSGL) and DSCL.

Table 6: Multi-stage geometry learning (MSGL) hyperparameters.

| Stage | Microphone Array | Number of channels | Learning Rate | Weight Decay | Epoch |
|-------|------------------|--------------------|--------------|--------------|-------|
| 1 | Tetrahedron (4 cm) | 4 | $2.5\times10^{-4}$ | $1.0\times10^{-4}$ | 1–10 |
| 2 | Dynamic | 4 | $5.0\times10^{-4}$ | $1.0\times10^{-6}$ | 11–20 |
| 3 | Dynamic | 4–12 | $1.0\times10^{-3}$ | $1.0\times10^{-6}$ | 21–300 |

Table 6 summarizes the MSGL setup. The training complexity was gradually increased: from fixed to dynamic MAs, and from a fixed to variable number of channels, with stage-specific learning rates and weight decay for stable convergence. Each epoch contained 28,800 utterances for training, and validation was conducted after every epoch (2,000 samples in stages 1–2, and 300 samples per channel in stage 3). DSCL employed multiple outputs with varying beamwidth targets, starting coarse and gradually refining them. Beamwidth parameters $\gamma_o$ were initialized as $[32°, 12°, 5°]$ and decreased linearly to $[5°, 5°, 5°]$ between epochs 35–60. This coarse-to-fine scheme first captures broad spatial patterns and then sharpens them into accurate spatial spectra for SSL.

## A.10 SYNTHETIC DATASET GENERATION

---

**Algorithm 3:** Synthetic dataset generation.

---

**Input:** Anechoic speech dataset, noise dataset, number of channels $C$
**Output:** Synthetic mixtures, oracle voice activity detection (VAD) labels, ground-truth DOAs
**foreach** *utterance* **do**

    **Speech selection:** Randomly select speakers and utterances, resample to 16 kHz, trim/pad to 4 s, and obtain oracle VAD labels using WebRTC VAD[a].

    **Noise generation:** Select one coherent noise, resample to 16 kHz, generate channel-wise Gaussian white noise, and trim/pad to 4 s.

    **Spatial setup:** Randomly configure room size, RT60, and $C$-channel 3D geometry with inter-microphone distances constrained by Eq. (23). Place speakers with at least 10° angular separation, with coherent noise placed at least 2.5 m away, and ensure the microphone array is at least 0.1 m from walls.

    **RIR generation:** Generate synthetic RIRs using gpuRIR (Diaz-Guerra et al., 2021b) with the image source method (Allen & Berkley, 1979) and diffuse modeling.

    **Signal construction:** Convolve signals with RIRs, mix speech with random SIR, mix noises with random SIR, and combine speech and noise with random SNR.

---

[a]https://github.com/wiseman/py-webrtcvad

$$R_{\min} = \left[\max\left(1, 4 - 3 \cdot \frac{C-4}{8}\right), 6\right] \text{ cm,}$$
$$R_{\max} = \left[7, \max\left(7, 9 + 4 \cdot \frac{C-4}{8}\right)\right] \text{ cm.}$$
(23)

This section details the process of generating the synthetic dataset, following the procedure of Baek et al. (2025). Algorithm 3 outlines the steps, while Eq. (23) defines the minimum $R_{\min}$ and maximum $R_{\max}$ inter-microphone distances as a function of the number of channels $C$. Table 1 summarizes the ranges of acoustic parameters used in Algorithm 3 for generating the synthetic dataset. Each parameter was randomly sampled from a uniform distribution within the specified interval, except for the elevation angle. The elevation was drawn from a von Mises-Fisher distribution (Fisher et al., 1993), given by $\frac{\varphi}{2}$, where $\varphi \sim \text{vMF}(\mu, \kappa)$ with $\mu = \pi$ and $\kappa = 2$, favoring positions near the horizontal plane while restricting the range to $[0, \pi]$.

For training and validation, anechoic utterances were taken from LibriSpeech (Panayotov et al., 2015), where the `train-clean-100` and `test-clean` sets were used, respectively, and noises were taken from MS-SNSD (Reddy et al., 2019), where the `train` and `test` sets were used, respectively. Anechoic speech for evaluation was taken from TIMIT (Garofolo et al., 1993), and noises from ESC-50 (Piczak, 2015).

### A.11 EVALUATION METRICS

MAE (°) and $\text{ACC}_{10}$ (%) are defined as:

$$\text{MAE}\,(°) = \frac{180}{\pi}\,\frac{1}{|\mathbf{L}_{\text{act}}|}\sum_{l\in\mathbf{L}_{\text{act}}}\frac{1}{T_l}\min_{p\in P_l}\sum_{s=1}^{T_l}\delta\left(\varsigma_s, \hat{\psi}_{p(s)}\right), \tag{24}$$

$$\text{ACC}_{10}(\%) = \frac{|\mathbf{L}_{\text{acc}}|}{|\mathbf{L}_{\text{act}}|}\times 100, \tag{25}$$

where $|\cdot|$ denotes the cardinality of a set, and $\mathbf{L}_{\text{act}}$ is the set of active speaker labels in the utterance. $T_l$ is the number of sources for the $l$-th frame, $P_l$ is the set of all permutations of $T_l$ sources, $\varsigma_s$ is the ground-truth DOA of source $s$, $\hat{\psi}_{p(s)}$ is the estimated DOA assigned to source $s$ under permutation $p$, and $\delta(\cdot)$ is the angular distance between two directions computed using Eq. (20). MAE computes the mean angular error between ground-truth and estimated DOAs, considering all possible source permutations to resolve permutation ambiguity in multi-source scenarios. $\mathbf{L}_{\text{acc}}$ is the subset of $\mathbf{L}_{\text{act}}$ whose angular error is less than $10°$.

### A.12 BOX AND VIOLIN PLOTS OF THE RESULTS

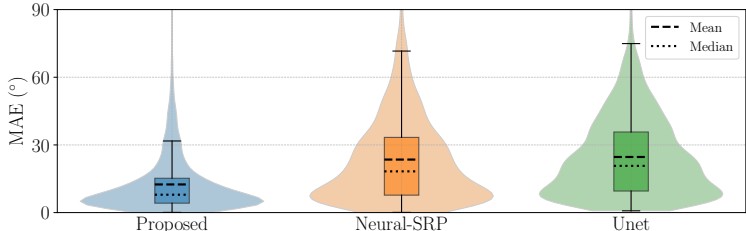

Figure 7: Box and violin plots of MAE for *Dynamic-S* with 4–12 channels.

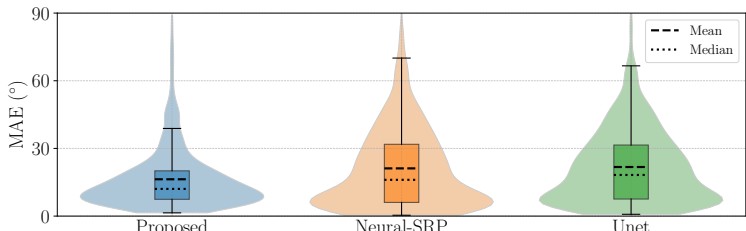

Figure 8: Box and violin plots of MAE for *Dynamic-U* with 13–16 channels.

Figures 7 and 8 present the box and violin plots of MAE results for *Dynamic-S* with 4–12 channels and *Dynamic-U* with 13–16 channels, respectively. Box plots show the interquartile range (IQR) and whiskers, where the black solid lines denote whiskers within 1.5 times the IQR. The dotted and dashed lines inside box plots indicate the median and mean of each method, respectively. Violin plots illustrate the kernel density estimation of the distribution. The proposed method exhibited a more pronounced peak around the median values with a narrower distribution, indicating that the results were more concentrated and consistent compared to the baselines, Unet and Neural-SRP, both equipped with AGG-RL.

## A.13 ANALYSIS ACROSS ENVIRONMENTAL CONDITIONS

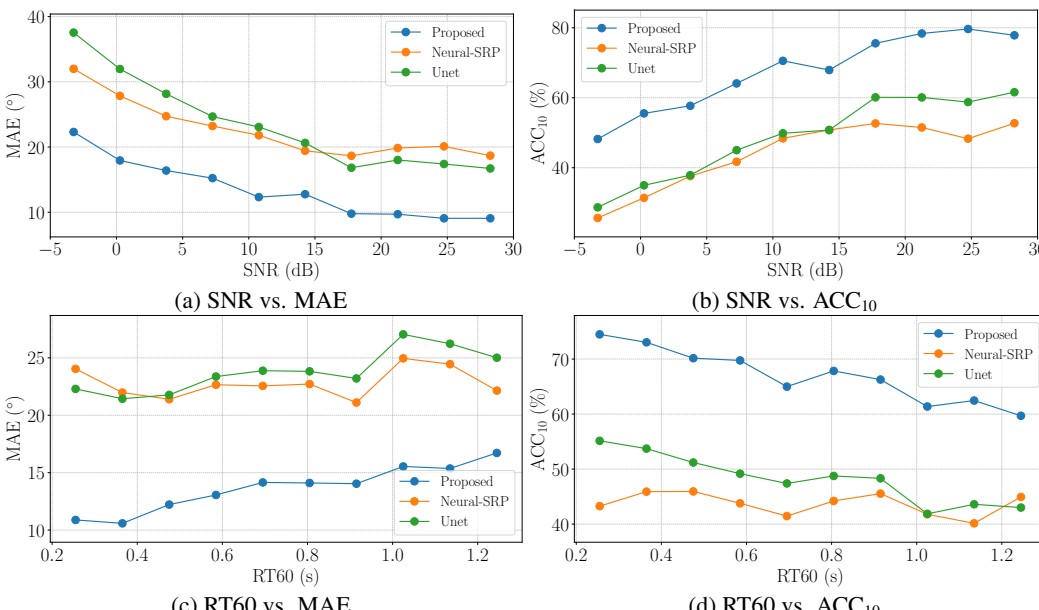

(a) SNR vs. MAE

(b) SNR vs. ACC$_{10}$

(c) RT60 vs. MAE

(d) RT60 vs. ACC$_{10}$

Figure 9: MAE and ACC$_{10}$ results of the proposed method and baselines (Unet, Neural-SRP; both with AGG-RL) on the *Dynamic-S* and *Dynamic-U* datasets across different SNR and RT60 conditions.

To assess the impact of environmental conditions on SSL performance, we analyzed the results across varying SNRs and RT60s using the synthetic datasets. Figure 9 illustrates the MAE and ACC$_{10}$ results for *Dynamic-S* and *Dynamic-U*, comparing the proposed method with the baselines (Unet and Neural-SRP), both equipped with AGG-RL. The two datasets were combined to provide a comprehensive overview of performance tendencies under varying environmental conditions. The combined datasets were divided into 10 equally sized bins for each SNR and RT60 range, and the nearest values to the bin centers were averaged. All methods showed improved performance with increasing SNR and decreasing RT60, as easier acoustic conditions inherently facilitate more accurate SSL. Notably, the proposed method consistently outperformed the baselines across all SNR and RT60 ranges, demonstrating strong robustness under both low-SNR and highly reverberant conditions.

Table 7: Experimental results under different numbers of active speakers across all datasets.

|  | *NAO robot* | | *Eigenmike* | | *Dynamic-S* | | *Dynamic-U* | |
|---|---|---|---|---|---|---|---|---|
|  | MAE | ACC$_{10}$ | MAE | ACC$_{10}$ | MAE | ACC$_{10}$ | MAE | ACC$_{10}$ |
| Unet |  |  |  |  |  |  |  |  |
| 1 speaker | 11.59 $\pm 3.43$ | 93.00 | 16.14 $\pm 3.69$ | 68.49 | 16.92 $\pm 0.91$ | 62.74 | 13.52 $\pm 1.20$ | 71.68 |
| 2 speakers | 19.78 $\pm 5.74$ | 24.41 | 23.29 $\pm 4.75$ | 12.14 | 32.46 $\pm 0.84$ | 29.76 | 30.06 $\pm 1.14$ | 33.98 |
| Neural-SRP |  |  |  |  |  |  |  |  |
| 1 speaker | 7.28 $\pm 2.62$ | 90.21 | 10.77 $\pm 2.28$ | 79.08 | 12.77 $\pm 0.84$ | 63.65 | 9.73 $\pm 0.97$ | 73.21 |
| 2 speakers | 13.28 $\pm 5.31$ | 33.74 | 17.89 $\pm 3.67$ | 17.24 | 33.55 $\pm 0.93$ | 20.92 | 31.22 $\pm 1.30$ | 26.47 |
| Proposed |  |  |  |  |  |  |  |  |
| 1 speaker | 7.48 $\pm 1.54$ | 96.06 | 9.77 $\pm 1.42$ | 81.13 | 7.08 $\pm 0.52$ | 85.95 | 9.51 $\pm 0.92$ | 74.09 |
| 2 speakers | 10.71 $\pm 3.54$ | 73.90 | 15.93 $\pm 3.56$ | 43.50 | 17.93 $\pm 0.73$ | 58.84 | 21.17 $\pm 1.10$ | 39.58 |

Table 7 summarizes the experimental results under different numbers of active speakers across all datasets. All methods showed degradation when moving from one to two speakers, which reflects the increased difficulty of SSL in multi-source scenarios. However, except for the MAE in the single-speaker condition of the *NAO robot* dataset—where Neural-SRP showed marginally better performance—the proposed method consistently outperformed both baselines across all datasets and speaker configurations, demonstrating its strong ability to handle both single- and multi-speaker spatial scenes.

## A.14 Results with STARSS23 Dataset

Table 8: Experimental results on *STARSS23* with different methods. The best results on each dataset are in **bold**.

| | Total | | 1 Speaker | | 2 Speakers | |
|---|---|---|---|---|---|---|
| | MAE | $ACC_{10}$ | MAE | $ACC_{10}$ | MAE | $ACC_{10}$ |
| Unet | 43.91 ±4.97 | 23.55 | 38.37 ±8.55 | 35.71 | 46.03 ±5.96 | 18.90 |
| Neural-SRP | 55.54 ±7.55 | 5.74 | 50.11 ±16.56 | 10.12 | 57.62 ±8.35 | 4.07 |
| Proposed | **27.32** ±5.50 | **36.09** | **18.67** ±6.15 | **56.58** | **30.63** ±6.96 | **28.25** |

For further validation, we evaluated the proposed method using the *STARSS23* dataset (Shimada et al., 2023), which consists of real recorded multi-speaker reverberant mixtures originally recorded with a 32-channel Eigenmike MA. The official dataset provides downmixed versions using 6-, 10-, 26-, and 22-th channel subsets, corresponding to a 4-channel tetrahedral-shaped configuration. Although the dataset was developed for sound event localization and detection (SELD), we focused exclusively on the speech localization aspect. Speech-only segments were selected based on the provided annotations. The results in Table 8 show a noticeable degradation compared to Table 3. This is expected because *STARSS23* involves low SNR conditions with various sound events, highly dynamic scenes with moving speakers, and acoustic characteristics that differ significantly from our static, synthetic training data. Despite these challenges, the proposed method consistently outperformed the baselines across all evaluation settings, demonstrating strong generalization capability to real-world and dynamic recording conditions.

## A.15 SPATIAL SPECTRUM VISUALIZATION

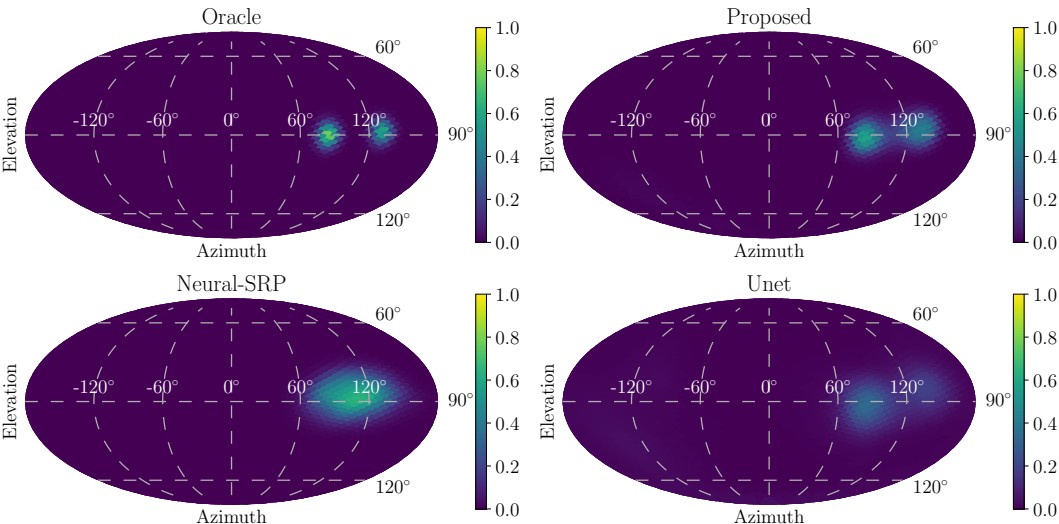

Figure 10: Spatial spectra averaged across frames for an utterance with two active speakers from the *NAO robot* dataset, visualized in Mollweide projection. Peaks correspond to estimated DOAs on the unit sphere.

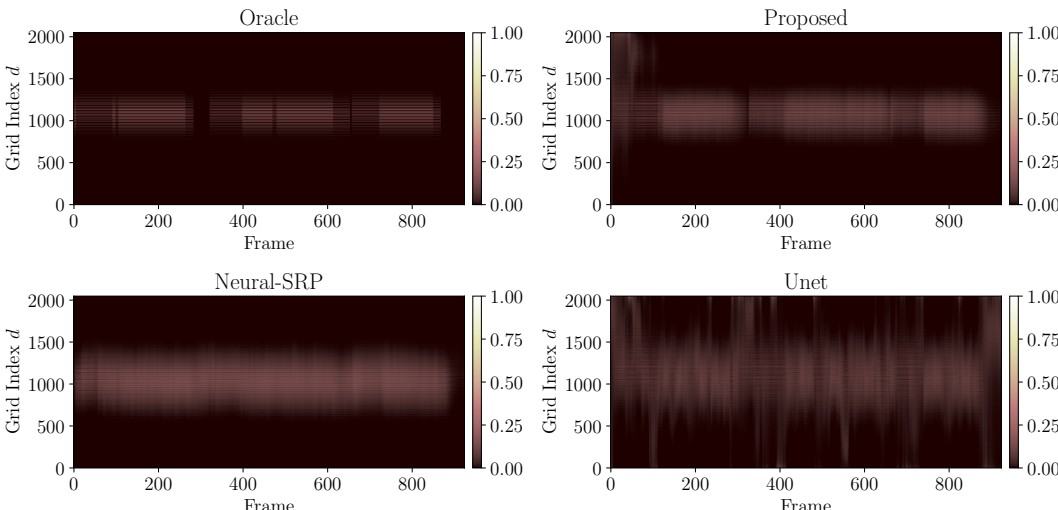

Figure 11: Frame-wise spatial spectra for the same utterance as Fig. 10, shown in 2D where the $x$-axis represents time frames and the $y$-axis denotes grid indices.

Figure 10 illustrates the spatial spectra of the oracle, the proposed method, Neural-SRP, and Unet on an utterance with two active speakers from the *NAO robot* dataset. For visualization, frame-level spectra were averaged and projected onto the Mollweide map. The proposed method produced sharp and distinct peaks exactly at both ground-truth DOAs, closely resembling the oracle. In contrast, Neural-SRP yielded a broad peak that failed to clearly separate the two speakers, while Unet generated less distinctive peaks. Figure 11 illustrates the spatial spectra in 2D, where $x$- and $y$-axes represent frame and DOA index $d$, respectively. The proposed method consistently produced stable peaks aligned with ground-truth DOAs and accurate VAD estimation, whereas Neural-SRP and Unet exhibited less consistent activations and unreliable VAD distinction.

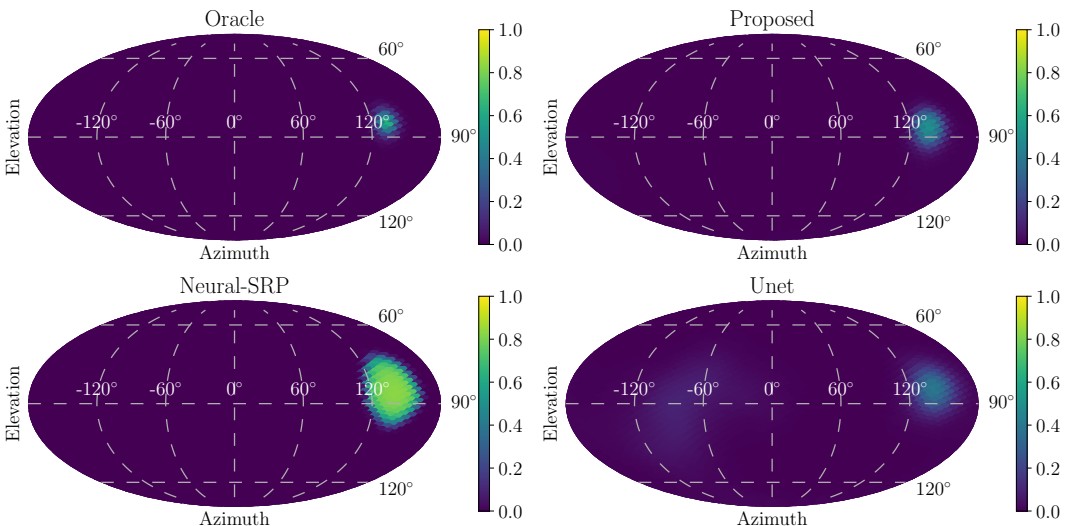

Figure 12: Spatial spectra averaged across frames for an utterance with a single active speaker from the *Eigenmike* dataset, visualized in Mollweide projection. Peaks correspond to estimated DOAs on the unit sphere.

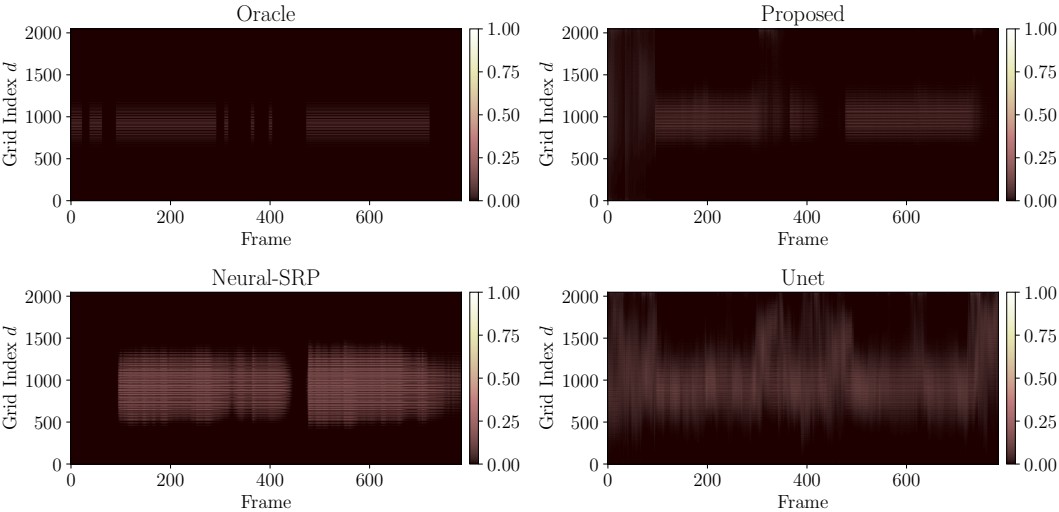

Figure 13: Frame-wise spatial spectra for the same utterance as Fig. 12, shown in 2D where the $x$-axis represents time frames and the $y$-axis denotes grid indices.

Figures 12 and 13 show the spatial spectra of the oracle, proposed method, Neural-SRP, and Unet on an utterance with a single active speaker from the *Eigenmike* dataset, using Mollweide projection and 2D frame-wise view, respectively. In the Mollweide projection (Fig. 12), the proposed method produced a distinct and sharp peak at the ground-truth DOA. By contrast, Neural-SRP yielded a broad peak that did not clearly indicate the source location, while Unet produced multiple non-ground-truth peaks (e.g., near -90° azimuth), demonstrating less reliable estimation. In the 2D representation (Fig. 13), the proposed method consistently localized the source with sharp peaks and reliable VAD, closely matching the oracle. Neural-SRP again exhibited a wide, ambiguous peak, and Unet generated less sharp peaks along with spurious responses. Overall, the proposed method yielded sharper and more stable spatial spectra than the baselines, highlighting its robustness for unseen MA geometries.

### A.16 Visualization of Grid Representations

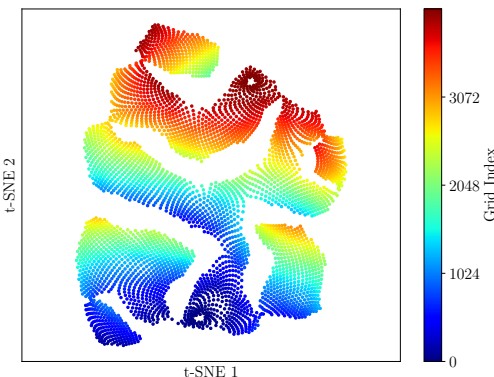

Figure 14: Grid representations visualized with t-SNE. Fibonacci grid points with $D = 4096$ are used as input candidates.

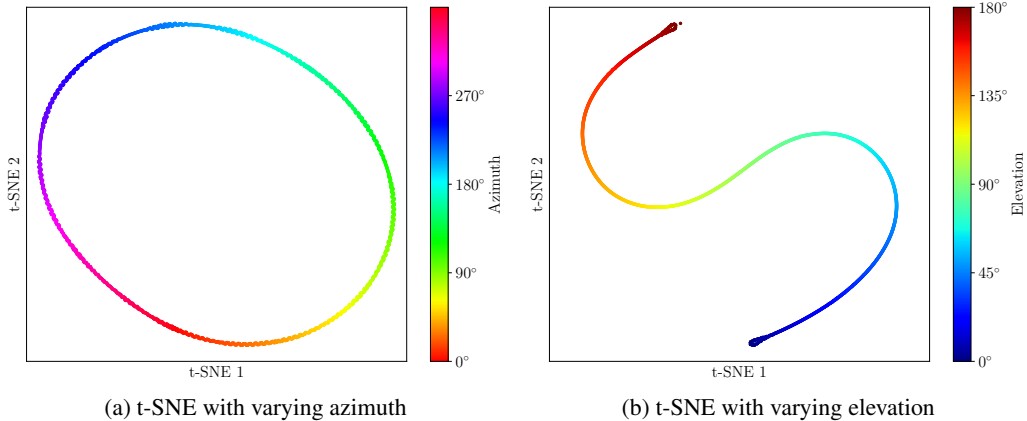

(a) t-SNE with varying azimuth         (b) t-SNE with varying elevation

Figure 15: Grid representations visualized with t-SNE for azimuth and elevation candidates.

Figure 14 visualizes the output with Fibonacci grid points with $D = 4096$. Nearby directions in 3D space were embedded closely, whereas opposite directions were mapped farther apart, demonstrating that the GRs preserved directional similarity in the latent space. Figure 15(a) illustrates the case of varying azimuth while fixing elevation at $90°$ with $D = 2048$. The t-SNE output formed a circular pattern consistent with azimuthal changes, indicating smooth representation of angular variation. Figure 15(b) shows the case of varying elevation while fixing azimuth at $0°$ with $D = 2048$. The t-SNE output formed an S-shaped pattern aligned with elevation changes, demonstrating that the Gridnet consistently encodes variations in elevation. These results indicate that Gridnet captured and preserved the relationships of candidate DOAs in the latent space, supporting flexible and interpretable candidate DOA representation.

### A.17 The Use of Large Language Models

The authors used large language models (LLMs) to assist in writing the manuscript, including refining language and ensuring clarity and coherence.

