# OpenReview forum: "Physics-Informed Audio-Geometry-Grid Representation Learning for Universal Sound Source Localization"
_ICLR.cc/2026/Conference — ICLR 2026 Poster_

### Official Review · Reviewer_ZXmA · 2025-10-30

**Soundness:** 3
**Presentation:** 3
**Contribution:** 3
**Rating:** 6
**Confidence:** 3

**Summary:**

The paper introduces the Audio-Geometry-Grid Representation Learning (AGG-RL) framework designed to achieve universal sound source localization (SSL) that is flexible regarding Direction-of-Arrival (DOA) grids and invariant to microphone array (MA) geometries. This is achieved by learning shared latent representations for audio-geometry information and DOA grid candidates, allowing the model to predict a probabilistic spatial spectrum without requiring retraining for new grid resolutions or MA configurations.
The primary technical contributions include two physics-informed components integrated into the framework: a Learnable Non-uniform Discrete Fourier Transform (LNuDFT), which adaptively allocates frequency bins to emphasize informative phase cues, and a relative Microphone Positional Encoding (rMPE), which encodes microphone coordinates relative to a reference channel, aligning with the nature of inter-channel time differences (TDOA). Experiments on synthetic and real datasets demonstrated that AGG-RL achieves better performance and stronger generalization to unseen microphone geometries and environments where traditional methods struggle.

**Strengths:**

- The paper integrates two novel, physics-informed components to enhance generalizability and interpretability. The LNuDFT optimizes frequency bin allocation to emphasize physically informative phase cues (IPDs) crucial for SSL. The rMPE encodes microphone coordinates relatively and improves generalization to unseen MA configurations.
- AGG-RL demonstrates better performance and robustness across experiments, showing strong generalization to unseen MA geometries, unseen numbers of channels (Dynamic-U), and real-world recordings.
- I like the visualization of the interpretable probabilistic spatial spectrum over candidate DOAs. It confirms that the proposed method yields sharper, more stable, and more distinct peaks closely aligned with ground-truth DOAs compared to baselines, even under unseen geometries.
- The paper is well-written, and I really enjoy reading it.

**Weaknesses:**

- The concept of the NuDFT itself is not new, having been explored in prior signal processing literature since the 1990s. The novelty lies in making the frequency bin locations $\nu_k$ learnable parameters within a deep neural network. While effective and physics-informed, this is an adaptation and optimization of an existing signal processing technique to maximize the extraction of IPDs.
- While the logit-based initialization strategy for LNuDFT proved effective in practice and benefited generalization to unseen conditions, the choice of mapping function and hyperparameters was made empirically

**Questions:**

- Could the authors elaborate on the empirical tuning process that led to the specific values chosen $\epsilon_{start}=0.15, \epsilon_{end}=0.95, \epsilon_{min}=0.01, \epsilon_{max} =100$, and how sensitive the final performance is to small variations in these constraints?
- While AGG-RL achieved good performance in general, the results still indicated a performance gap between seen and unseen conditions. What specific architecture or training configurations do the authors believe offer the most potential to further minimize this performance gap under diverse, unseen MA geometries?
-  In the ablation study, experiment (vi) (Fixed Grid) achieved the best performance on the Dynamic-S dataset (seen training conditions), and surprisingly, slightly outperformed the proposed AGG-RL method on the Dynamic-U dataset (unseen channels) in terms of ACC10. Why did the AGG-RL framework exhibit a degradation compared to the fixed-grid approach in this specific unseen synthetic channel configuration?

---

> ### Author Response · Authors · 2025-11-20
> **Response to Reviewer ZXmA (1/2)**
>
> We sincerely thank the reviewer for the thoughtful and constructive feedback. Below, we provide detailed responses to all comments regarding the identified weaknesses and questions. For items that led to changes in the manuscript, we indicate the updated line numbers or sections for clarity.
>
> ---
>
> **W1. The concept of the NuDFT itself is not new, having been explored in prior signal processing literature since the 1990s. The novelty lies in making the frequency bin locations  learnable parameters within a deep neural network. While effective and physics-informed, this is an adaptation and optimization of an existing signal processing technique to maximize the extraction of IPDs.**
>
> * We fully agree that NuDFT has a long history in classical signal processing, and the manuscript already includes citations that acknowledge this prior work. However, to the best of our knowledge, this is the first study to integrate NuDFT with learnable frequency-bin locations into a deep neural architecture for spatial audio tasks, particularly sound source localization (SSL). This integration is non-trivial for two reasons:
>
>   1. Task-specific adaptation: SSL relies critically on informative IPD regions. Making the bin locations learnable allows the model to allocate higher resolution to physically meaningful mid-frequency bins, in contrast to the fixed uniform spacing used in the standard DFT. This yields a task-oriented spectral representation tailored specifically for SSL.
>   2. Stable training via constrained parameterization: We designed LNuDFT with monotonicity and Nyquist constraints to preserve physical validity, and introduced a logit-based initialization that biases learning toward acoustically relevant frequency allocations. These components were crucial for stable convergence and effective learning within an end-to-end neural pipeline. We believe these design choices constitute a novel contribution that enables practical and physically grounded integration of NuDFT into deep learning frameworks, with potential applicability beyond SSL.
>
>   Therefore, while NuDFT itself is not new, our contribution lies in a trainable, physics-informed, and SSL-oriented formulation that differs fundamentally from prior handcrafted strategies with fixed parameters.
>
> ---
>
> **W2., Q1. While the logit-based initialization strategy for LNuDFT proved effective in practice and benefited generalization to unseen conditions, the choice of mapping function and hyperparameters was made empirically. Could the authors elaborate on the empirical tuning process that led to the specific values chosen $\epsilon_{start}=0.15,\epsilon_{end}=0.95,\epsilon_{min}=0.01, \epsilon_{max}=100$, and how sensitive the final performance is to small variations in these constraints?**
>
> * First, we experimented with uniform initialization (equivalent to the standard DFT) and observed that the model converged toward denser sampling in the mid-frequency range. This aligns with the well-known fact that mid-frequency IPD cues are the most reliable and informative for SSL tasks. Based on this observation, we designed the logit-based initialization to bias the frequency allocation toward mid frequencies while still allowing adaptation through learnable parameters.
> * The selected values of $\epsilon_{start}=0.15$ and $\epsilon_{end}=0.95$ were chosen to assign denser sampling to the low- and mid-frequency regions while sparsely covering the high-frequency region to avoid spatial aliasing. As shown in Fig. 3, this initialization produces frequency allocations that concentrate primarily on mid-frequency bands where IPD cues are most reliable. Furthermore, the initialization also assigned slightly denser sampling in the low-frequency bands than in the high-frequency bands, ensuring stable IPD information while avoiding aliasing effects. We conducted a grid search over $\epsilon_{start}$ and $\epsilon_{end}$ and found that small perturbations around these values did not significantly affect performance. The chosen configuration yielded the most consistent and robust results across our experiments. This clarification has been added in Lines 910–917 of the revised manuscript.
> * The bounds $\epsilon_{min}=0.01$ and $\epsilon_{max}=100$ served as safeguards to prevent extreme values that could destabilize training. In practice, the learnable parameters $a_{k}$ did not approach these bounds, but we included them to ensure numerical stability and monotonicity in all training conditions, while allowing sufficient flexibility.
>
> ---

---

> ### Author Response · Authors · 2025-11-20
> **Response to Reviewer ZXmA (2/2)**
>
> ---
>
> **Q2. While AGG-RL achieved good performance in general, the results still indicated a performance gap between seen and unseen conditions. What specific architecture or training configurations do the authors believe offer the most potential to further minimize this performance gap under diverse, unseen MA geometries?**
>
> * First, few-shot fine-tuning represents a straightforward and highly effective strategy for reducing the performance gap under unseen geometries. Because AGG-RL already provides a strong geometry-agnostic initialization, adapting only a small subset of parameters using a small amount of geometry-specific data would directly improve generalization to new microphone arrays.
> * Second, test-time adaptation (TTA) techniques could further reduce mismatches by adjusting model statistics or selected parameters during inference. Such approaches would allow AGG-RL to adapt dynamically to unseen geometries or acoustic conditions without requiring explicit offline fine-tuning.
> * Third, improving the multi-head self-attention (MHSA) mechanism—which is responsible for modeling inter-microphone relationships—could enhance length generalization when the number of microphones changes. More advanced attention architectures may provide stronger invariance to diverse array configurations.
>
>   These directions could further strengthen AGG-RL’s robustness under previously unseen microphone array geometries, and represent important avenues for future research.
>
> ---
>
> **Q3. In the ablation study, experiment (vi) (Fixed Grid) achieved the best performance on the Dynamic-S dataset (seen training conditions), and surprisingly, slightly outperformed the proposed AGG-RL method on the Dynamic-U dataset (unseen channels) in terms of ACC10. Why did the AGG-RL framework exhibit a degradation compared to the fixed-grid approach in this specific unseen synthetic channel configuration?**
>
> * Experiment (vi) shared the same front-end (LNuDFT with logit-based initialization and GCC-PHAT with rMPE) as the proposed method, but differed in its output representation: it directly predicted scores on a fixed discrete grid.
>
>   Under the _Dynamic-U_ dataset settings, it achieved a slightly higher ACC10 than AGG-RL. This behavior can be explained by the fact that experiment (vi)’s fixed-grid training objective was aligned with the fixed candidate directions used for evaluation. In such conditions, this alignment allowed experiment (vi) to benefit from a tightly matched configuration. In contrast, AGG-RL was designed to generalize across arbitrary candidate grids rather than optimizing for a single fixed grid. This broader objective introduced a small trade-off in _Dynamic-U_ but yielded substantial gains in real-world settings, as evidenced by the significantly better performance on the _Eigenmike_ dataset.
>
> * Overall, the slight advantage of experiment (vi) in _Dynamic-U_ stems from its fixed-grid training being perfectly aligned with the evaluation grid used in this specific synthetic setup. In contrast, AGG-RL is explicitly designed to remain grid-flexible and to generalize across arbitrary candidate grids—an ability that becomes crucial in real-world conditions, where it consistently outperformed experiment (vi).
> ---

---

### Official Review · Reviewer_LRbg · 2025-11-01

**Soundness:** 2
**Presentation:** 3
**Contribution:** 2
**Rating:** 4
**Confidence:** 3

**Summary:**

This paper presents AGG-RL (Audio Geometry Grid Representation Learning), a family of algorithms to estimate direction-of-arrival in a way that is invariant to the geometry of the microphone array and adaptable to different grids.

To achieve this, AGG-RL relies on two innovations: i) using a DFT where the width of the frequency bins is learnt alongside the training (allowing for higher resolution on the frequencies that are most relevant for direction-of-arrival estimation), and ii) use of relative position encoding to describe the microphone geometry.

To train AGG-RL a network (AuGeonet)  encodes the sound field (conditioned on the microphone array geometry), whereas another network (Gridnet) encodes a grid of query points where the sources in the sound-field may be localised. The final output logits are obtained by combining the outputs of AuGeonet and Gridnet (essentially an inner product), with each logic representing the probability of a source being present at the corresponding query point. Binary cross-entropy is used to compute the loss.

Evaluations on the LOCATA dataset show AGG-RL achieving ~11° mean-absolute error on direction-of-arrival estimation on a microphone array and dataset not seen during training.

**Strengths:**

* The paper is well written and easy to follow (though see nitpicks below for possible improvements).
* The contributions (learnable DFT widths, relative positions for microphone array embeddings, and flexible grids) are elegant, inspired by physics, and well described in the manuscript.
* The paper contains a useful introduction to direction-of-arrival estimation, allowing non-acoustics-experts to follow the paper.
* The evaluation ablates all the components of AGG-RL and presents very strong results.
* Device-independent direction-of-arrival estimation is an important problem for acoustics: many consumer devices nowadays need to perform direction-of-arrival estimation, but each tends to have their own configuration. A single network that generalises over all devices is genuinely useful.

**Weaknesses:**

* **W1**: Some details of the training and evaluation procedures are missing in the manuscript (see questions below)
* **W2**: More ablations are needed to fully characterise the behaviour of AGG-RL (eg. SNR, number of sources in the sound-field, see questions below for specifics).
* **W3**: Evaluating against other datasets with different microphone array geometries would help definitely establish generalisation to any microphone array.

_Overall_, this is a strong submission held back by a relatively short evaluation that leaves many questions unanswered. It is my hope many of these issues are resolved during rebuttal.

**Questions:**

* **Q1**: In eq 14, are the grid points $\\theta_d$ and $\\phi_d$ obtained by linearly spacing the azimuth and elevation ranges by $\\frac{G}{2}$ points respectively?
* **Q2**: How many synthetic samples were used during training? Can the performance of AGG-RL be further improved by adding further training samples?
* **Q3** Appendix A.9 contains really important information that it is not hinted at in the main text (acoustics characteristics of the simulations, utterances and noise datasets, SNRs, duration of the audio segments) that should be summarised in the main text.
* **Q4**: What does it mean exactly that the NAO dataset has training exposure? Was the dataset used during training?
* **Q5**: In table 2, could you report the spread of the mean-absolute-error of the direction-of-arrival errors?
* **Q6**:  How do you train AuGeonet to accept more/fewer microphones in an array? Is there a "not present" encoding?
* **Q7**: Could you breakdown the results in table 2 by number of speakers present in the sound-field?
* **Q8**: Could you breakdown the results in table 2 by SNR and T60?
*  **Q9**: Can AGG-RL model different directional sensitivities in microphones? If not, could you hypothesise if this would be possible with an extension?  Relatedly, would it be possible to input ideal ambisonics to AGG-RL?
* **Q10**: What microphone array geometry information is provided to AuGeonet? The main manuscript seems to indicate it is the relative encodings just the relative angles, but Appendix A.3 indicates the geometry is given in cartesian and polar coordinates.
* **Q11**: Could you evaluate AGG-RL against other datasets for direction-of-arrival estimation in the literature (eg. TUT Sound Events 2018[1], Spatial LibriSpeech [2], STARSS'23 [3]?
* **Q12**: In table 11, why does the performance decay as the number of points in the grid increases.


-----

### Nitpicks (do not affect rating, no need to follow up on these during rebuttal)

* **N1**: I found the notation for $\bf{v}$ very confusing and for a few minutes could not understand what $\bf{v}_M$ and $\bf{v}_G$ were. These would be easier to parse if they were expressed as a function $v(Q) = \frac{4}{Q}\[0, 1, \dots, \frac{Q}{4} -1\]$
* **N2**: Similarly I find it would be easier if you indicated the function parameters in  equations 12, 14,15 and 16.
* **N3**: I would suggest explicitly mentioning in equation 1 that $c, k, l$ refer to the channel, frequency, and time indices respectively.


-------
[1] Adavanne et al. (2019) “Sound Event Localization and Detection of Overlapping Sources Using Convolutional Recurrent Neural
Networks”. Journal of Selected Topics in Signal Processing.

[2] Sarabia et al. (2023) “Spatial LibriSpeech: An Augmented Dataset for Spatial Audio Learning” Interspeech.

[3] Shimada et al. (2023) "STARSS23: An Audio-Visual Dataset of Spatial Recordings of Real Scenes with Spatiotemporal Annotations of Sound Events" NeurIPS Track on Datasets and Benchmarks.

---

> ### Author Response · Authors · 2025-11-20
> **Response to Reviewer LRbg (1/3)**
>
> We sincerely thank the reviewer for the thoughtful and constructive feedback. Below, we provide detailed responses to all comments. Since several weaknesses were phrased as questions, we address them together in the corresponding responses. We have also incorporated the suggested nitpick revisions. For items that led to changes in the manuscript, we indicate the updated line numbers or sections for clarity.
>
> ---
>
> **Q1. In eq 14, are the grid points $\theta_{d}$ and $\phi_{d}$ obtained by linearly spacing the azimuth and elevation ranges by $\tfrac{G}{2}$ points respectively?**
>
> * We used Fibonacci sphere sampling to generate the $D$ grid points, which produces an approximately uniform distribution over the sphere (depicted in Lines 301–302 and Appendix A.5 in the revised manuscript). Thus, the grid points are not obtained by linearly spacing azimuth and elevation independently.
>
> ---
>
> **Q2. How many synthetic samples were used during training? Can the performance of AGG-RL be further improved by adding further training samples?**
>
> * Each training epoch consisted of 28,800 synthetic samples (Line 1097), and the samples were generated on-the-fly with dynamic RIRs to ensure high diversity. Increasing the number of training epochs—effectively exposing the model to more synthetic samples—did not yield further improvements and sometimes led to mild overfitting. These observations indicate that the model was already sufficiently trained under the current setup, and simply adding more synthetic samples did not provide additional performance gains.
>
> ---
>
> **Q3. Appendix A.9 contains really important information that it is not hinted at in the main text (acoustics characteristics of the simulations, utterances and noise datasets, SNRs, duration of the audio segments) that should be summarised in the main text.**
>
> * Following the reviewer’s suggestion, we have incorporated a concise summary of the key acoustic characteristics into the main text. Specifically, Table 1 in the revised manuscript now provides an overview of the simulation configurations (e.g., SNRs, RT60s, etc.). In addition, Lines 337–357 and 373 summarize the RIR generation process along with the speech and noise datasets used during data simulation.
>
> ---
>
> **Q4. What does it mean exactly that the NAO dataset has training exposure? Was the dataset used during training?**
>
> * “Training exposure” in Table 2 means that the number of channels in the _NAO robot_ dataset (i.e., 12 channels) was included in the synthetic training configurations. However, the _NAO robot_ recordings themselves were not used during training; only synthetic data with the same channel count were generated to expose the model to that geometry.
>
> ---
>
> **Q5. In table 2, could you report the spread of the mean-absolute-error of the direction-of-arrival errors?**
>
> * We have added 95% confidence intervals (CIs) for the mean absolute error (MAE) in Tables 3, 4, 7, and 8 of the revised manuscript. These CIs quantify the spread of the MAE values as requested and help clarify the statistical significance of the performance differences observed across methods.
>
> ---
>
> **Q6. How do you train AuGeonet to accept more/fewer microphones in an array? Is there a "not present" encoding?**
>
> * AuGeonet is designed following the same principle as GI-DOAEnet, namely, that it should not rely on a fixed number of microphones. Instead, it processes each microphone pair independently via GCC-PHAT and aggregates the resulting pairwise features in a permutation-invariant manner. Because the model operates on a variable-length set of microphone-pair features, it inherently supports arbitrary numbers of microphones without requiring any form of explicit masking or “not present” encoding.
>
>   In addition, the multi-head self-attention (MHSA) applied along the channel dimension further enables the model to learn inter-microphone relationships regardless of array size, ensuring robustness to changes in the number of microphones.
>
> ---
>
> **Q7. Could you breakdown the results in table 2 by number of speakers present in the sound-field?**
>
> * As requested, we have added a detailed breakdown of the results by the number of active speakers. Specifically, the revised manuscript now includes Table 7 in Appendix A.13, which reports the performance separately for one-speaker and two-speaker conditions corresponding to Table 3 in the main text.
>
>   The results show that performance decreased as the number of active speakers increases, which is expected due to the increased difficulty of the localization task in multi-speaker mixtures. Importantly, the proposed method consistently outperformed both baselines across almost all datasets and speaker configurations, except for the MAE in the single-speaker condition of the _NAO robot_ dataset, where Neural-SRP showed marginally better performance, confirming its robustness in more challenging acoustic scenes.
>
> ---

---

> ### Author Response · Authors · 2025-11-20
> **Response to Reviewer LRbg (2/3)**
>
> ---
>
> **Q8. Could you breakdown the results in table 2 by SNR and T60?**
> * As requested, we have added a detailed breakdown of the results by SNR and RT60. The revised manuscript now includes Fig. 9 in Appendix A.13, which visualizes the performance trends corresponding to Table 3 in the main text.
>
>   The analysis shows that performance degraded as SNR decreased and RT60 increased, which is expected because noise and reverberation reduce the reliability of phase-based spatial cues. Importantly, the proposed method remained consistently more robust than the baselines across all SNR and RT60 ranges, demonstrating strong generalization under challenging acoustic conditions.
> ---
> **Q9. Can AGG-RL model different directional sensitivities in microphones? If not, could you hypothesise if this would be possible with an extension? Relatedly, would it be possible to input ideal ambisonics to AGG-RL?**
> * Directional sensitivities.
>
>   The current AGG-RL architecture assumes omnidirectional microphones and does not explicitly model directional sensitivities. However, it could naturally be extended to incorporate directional patterns by training with data that simulates such characteristics. In practice, the real-world microphone arrays used in our experiments—although nominally omnidirectional, they can exhibit mild structural directivity due to their spherical housing and port geometry (e.g., the NAO robot and the Eigenmike). AGG-RL generalized well to such cases, suggesting that the model can implicitly accommodate moderate directivity variations.
> * Ambisonics inputs.
>
>   AGG-RL explicitly needs the microphone coordinates to handle arbitrary array geometries. Therefore, it is not possible to directly input ideal ambisonics to AGG-RL. However, if appropriate transformations are applied to convert ambisonics signals into equivalent microphone signals at known positions, AGG-RL could then process these transformed signals effectively.
> ---
> **Q10. What microphone array geometry information is provided to AuGeonet? The main manuscript seems to indicate it is the relative encodings just the relative angles, but Appendix A.3 indicates the geometry is given in cartesian and polar coordinates.**
> * The microphone array geometry is initially provided in Cartesian coordinates. From these Cartesian coordinates, the pairwise relative positions between microphones are computed, and these positions are then converted into spherical coordinates (azimuth, elevation, and distance).
>
>   In the AuGeonet, both representations are used as follows:
>
>   * Relative Cartesian coordinates and their spherical counterparts are concatenated with the input of the initial ConvBlock, as described in Line 955, Appendix A.3 of the revised manuscript.
>   * Spherical coordinates are further used to compute the rMPE features (Eq. (12) in the main manuscript).
>
>   In summary, the geometry is provided in Cartesian coordinates, and both relative Cartesian and spherical coordinates are computed and utilized within the model.
> ---

---

> ### Author Response · Authors · 2025-11-20
> **Response to Reviewer LRbg (3/3)**
>
> ---
> **Q11. Could you evaluate AGG-RL against other datasets for direction-of-arrival estimation in the literature (eg. TUT Sound Events 2018[1], Spatial LibriSpeech [2], STARSS'23 [3])?**
> * _TUT Sound Events 2018_ and _Spatial LibriSpeech_ mainly provide ambisonic (first-order spherical harmonic) recordings or 2D circular microphone arrays, which are not aligned with the focus of our work—generalization across arbitrary 3D microphone array geometries. Since our method explicitly leverages microphone geometry (via rMPE), ambisonic-format datasets are not suitable for a fair evaluation.
>
>   In contrast, _STARSS23_ includes real-world recordings using a 4-channel tetrahedral array derived from a 32-channel Eigenmike. This format directly matches the geometry-aware setting required for evaluating AGG-RL. Therefore, we selected _STARSS23_ as a representative external benchmark.
> * We conducted experiments on _STARSS23_ and reported the results in Appendix A.14 (Table 8) of the revised manuscript. A summary is also shown below. Although the absolute performance differs due to dataset difficulty (low SNR, moving speakers), the proposed AGG-RL consistently outperformed the baselines, demonstrating strong generalization capability.
> ### Experimental results on STARSS23 with different methods
> |  | Total | Total | 1 Speaker  | 1 Speaker  | 2 Speakers   | 2 Speakers|
> |:-----------:|:---------:|:-----------:|:---------:|:-----------:|:---------:|:-----------:|
> |                    | MAE                                                   | ACC10                                                 | MAE                     | ACC10           | MAE                     | ACC10           |
> | Unet               | 43.91 ± 4.97                                         | 23.55                                                 | 38.37 ± 8.55           | 35.71           | 46.03 ± 5.96           | 18.90           |
> | Neural-SRP         | 55.54 ± 7.55                                         | 5.74                                                  | 50.11 ± 16.56          | 10.12           | 57.62 ± 8.35           | 4.07            |
> | **Proposed** | **27.32** ± 5.50                               | **36.09**                                       | **18.67** ± 6.15 | **56.58** | **30.63** ± 6.96 | **28.25** |
> ---
>
> **Q12. In table 11, why does the performance decay as the number of points in the grid increases?**
> * We believe the reviewer is referring to Table 4 in the revised manuscript. As shown in Table 4, which evaluated different numbers of grid points $D$, the performance generally improved as $D$ increased and stabilizes around $D \geq 512$. For the synthetic datasets, larger $D$ provided slightly better results because finer angular discretization reduced quantization errors under simulated (i.e., matched) conditions.
>
>   In contrast, the real datasets showed saturation or mild degradation beyond $D=2048$. We attribute this to the fact that overly dense candidate grids made the localization decision more sensitive to small acoustic mismatches inevitably present in real recordings, such as sensor noise, array manufacturing tolerances, and synthetic RIR deviations. These observations and interpretations have been added in Lines 475–487 of the revised manuscript.
>
>
> ---
> **N1. I found the notation for $\mathbf{v}$ very confusing and for a few minutes could not understand what $\mathbf{v}_M$ and $\mathbf{v}_G$ were.** **These would be easier to parse if they were expressed as a function $v(Q)=\tfrac{4}{Q}[0,1, \ldots, \tfrac{Q}{4}-1]$.**
>
> * We have revised the notation accordingly in Eqs. (11), (12), and (14) of the revised manuscript to improve clarity.
> ---
> **N2. Similarly I find it would be easier if you indicated the function parameters in equations 12, 14,15 and 16.**
> * We have modified Eqs. (12), (13), (14), and (15) to explicitly include their function parameters for better readability.
> ---
> **N3. I would suggest explicitly mentioning in equation 1 that $c,k,l$ refer to the channel, frequency, and time indices respectively.**
> * These notations are now explicitly defined in Lines 184–185 of the revised manuscript, as suggested.
> ---

---

> > ### Comment · Reviewer_LRbg · 2025-11-26
> >
> > Thank you for your detailed responses to all of my questions. I appreciate there were quite a few of them.
> >
> > I am mostly convinced by all your clarifications and arguments, and I will definitely revise my rating upwards.
> >
> > However, there is one sticking point that would change the final rating in my mind, and it is this: the performance on STARSS'23 (across MAE, ACC$_{10}$ for 1 and 2 speakers) is significantly worse than the baselines presented in Tables 3 and 4 (in the revised manuscript). Eg. 18.67° (on STARSS'23 1speaker) vs 11.24° (on Eigenmike).
> >
> > Inspired by your response to reviewer ZXmA, could I ask you to fine-tune the model on STARSS'23 to verify that the performance gets closer to the validation datasets? This would alleviate concerns over generalisation.

---

> > > ### Author Response · Authors · 2025-11-28
> > > **Response to Reviewer LRbg (1/1)**
> > >
> > > We sincerely thank the reviewer for the positive feedback and for kindly reconsidering the rating. To address your remaining concern regarding the generalization gap on _STARSS23_, we conducted a lightweight fine-tuning experiment exactly as suggested.
> > >
> > > ---
> > >
> > > We used the following experimental setup for the fine-tuning:
> > >
> > > * Dataset: STARSS23 dev-train split (61 audio samples)
> > > * Preprocessing: randomly cropped/padded to 32 seconds segments, batch size = 1
> > > * Training: 80 epochs, ~7 seconds per epoch → ~9.3 minutes total
> > > * Optimizer: Adam, learning rate = 1e-4
> > > * Initialization: the proposed pretrained AGG-RL model
> > >
> > > ---
> > >
> > > The results on the _STARSS23_ dev-test split after fine-tuning are summarized in the table below, along with the original results for comparison:
> > >
> > >
> > > ### Experimental results on _STARSS23_ with different methods
> > > |  | Total | Total | 1 Speaker  | 1 Speaker  | 2 Speakers   | 2 Speakers|
> > > |:-----------:|:---------:|:-----------:|:---------:|:-----------:|:---------:|:-----------:|
> > > |                    | MAE                                                   | ACC$_{10}$                                                | MAE                     | ACC$_{10}$           | MAE                     | ACC$_{10}$           |
> > > | Unet               | 43.91 ± 4.97                                         | 23.55                                                 | 38.37 ± 8.55           | 35.71           | 46.03 ± 5.96           | 18.90           |
> > > | Neural-SRP         | 55.54 ± 7.55                                         | 5.74                                                  | 50.11 ± 16.56          | 10.12           | 57.62 ± 8.35           | 4.07            |
> > > | Proposed | 27.32 ± 5.50                               | 36.09                                       | 18.67 ± 6.15 | 56.58 | 30.63 ± 6.96 | 28.25 |
> > > Fine tuned | **24.24** ± 5.50 | **39.63** | **16.18** ±  4.62 | **57.43** | **27.33** ± 7.43 | **34.36**
> > >
> > > These improvements confirm that AGG-RL can adapt to new acoustic environments with a small amount of real-world data.
> > >
> > > ---
> > >
> > > Although both _STARSS23_ and _Eigenmike_ were real-recorded datasets, key differences explain the performance gap:
> > >
> > > * Microphone Array Size: _Eigenmike_ used a 32-channel spherical array, while _STARSS23_ employed a 4-channel tetrahedral array, resulting in substantially weaker spatial resolution.
> > > * Speaker Motion: AGG-RL was trained on synthetic mixtures with static speakers, while _STARSS23_ included moving speakers, making SSL more challenging.
> > > * Interfering sound events: _STARSS23_ contained high-power sound events treated as noise in this task, creating lower-SNR conditions and more ambiguous IPD cues.
> > >
> > > These differences introduced significant domain shifts that challenged the generalization of the pretrained AGG-RL model. Nevertheless, the proposed method already outperformed both baselines without fine-tuning, and the fine-tuned version achieved further gains. This additional experiment demonstrates that AGG-RL can adapt to new acoustic conditions with limited data, supporting its practical generalization capability. Exploring more systematic domain-adaptation strategies to further enhance robustness across diverse real-world settings would be a promising avenue for future work.
> > >
> > > ---
> > >
> > >
> > >
> > >
> > > We sincerely appreciate the reviewer’s suggestion, which helped strengthen our empirical validation. We hope this resolves the remaining concern.

---

### Official Review · Reviewer_aRq1 · 2025-11-07

**Soundness:** 3
**Presentation:** 3
**Contribution:** 3
**Rating:** 6
**Confidence:** 3

**Summary:**

This paper introduces AGG-RL, a novel framework for sound source localization (SSL). It aims to achieve SSL with geometric invariance and grid flexibility by jointly learning audio-geometric representations and grid representations in a shared latent space. To achieve this, the approach proposes a learnable non-uniform discrete Fourier transform (LNuDFT) that assigns frequency bins based on physical informativeness, and a relative microphone position encoding (rMPE) aligned with TDOA. Experiments were conducted on synthetic and real datasets, demonstrating improved performance over baselines.

**Strengths:**

- **Originality**: The integration of physics-informed design (LNuDFT, rMPE) with a unified latent space for audio, geometry, and grid is original and is well-motivated by known limitations of geometry-specific and fixed-grid methods in SSL.
- **Quality**: The experimental design is thorough: ablation studies, varied datasets (including real recordings), and consistent baselines provide credible evidence for claims of generalization and robustness. The efficiency analysis (parameter count, FLOPs) is also useful.
- **Clarity**: The paper is mostly well written, with helpful architectural diagrams and clear mathematical formulations. The method and its motivation are explained coherently, and appendices include implementation details and sample code links.
- **Significance**: The experimental results demonstrate that this framework provides substantial improvements to SSL in real-world environments where array and grid conditions may vary.

**Weaknesses:**

1. **(Lack of justification for the claim on being physics-informed)** The paper mentions LNuDFT and rMPE as two main components proposed as ‘physics-informed components’ in the abstract, but the reason why these are ‘physics-informed’ was not explained.
    - Although Appendix A.1 supports that LNuDFT's ‘trainable manner’ emphasizes informative phase regions/causes to some extent, but the way of drawing the claim that 'it emphasizes informative regions' was limited to the common sense about speech signals. (For example, would the frequency response of LNuDFT parameters become different for the other signals, e.g., bird chirps or ambient wind noise etc?)
    - Similarly, for rMPE, it was not demonstrated that using this method actually captures “inter-channel time differences” more effectively.

    Performing DFT non-uniformly and encoding information relatively can be done independently of physics; connecting it to physical phenomena seems to require more detailed justification.

2. **(Difficulty in result analysis)** While it is commendable that comparisons were made on both real and synthetic datasets, the fact that all microphone arrays in the synthetic data were selected as dynamic microphones complicates result analysis.
    - Particularly, examining the results in Table 3 reveals a noticeable trend of performance differences, but it is unclear whether this stems from differences between real and synthetic data or from differences in dynamic microphones. The authors also did not draw a clear conclusion on this point.

**Questions:**

1. Experiment (v) still outperforms the proposed one on the Dynamic-U data and performs on par with the proposed one for NAO robot, but significantly degrades on the Eigenmike. Please expand on this discrepancy?

2. In L70, “Both components facilitate the extraction of spatial representations with physics-based inductive biases”: Why are LNuDFT and rMPE considered as imposing (physics-based) inductive biases? To me, these are perceived as a process that introduces and makes learnable parameters for feature extraction more flexible, and I see it as a process that relieves the inductive bias.

3. Can the LNuDFT initialization or update scheme get stuck in poor local minima, e.g., if the initial frequency allocation is far from optimal? Have the authors tried more physically motivated or data-driven initializations beyond logit-based mapping?

---

> ### Author Response · Authors · 2025-11-20
> **Response to Reviewer aRq1 (1/2)**
>
> We sincerely thank the reviewer for the thoughtful and constructive feedback. Below, we provide detailed responses to all comments, including both weaknesses and questions. For items that led to changes in the revised manuscript, we indicate the corresponding line numbers or sections for clarity.
>
> ---
> **W1. Lack of justification for the claim on being physics-informed: LNuDFT and rMPE.**
> * LNuDFT.
>
>   In our experiments, the target signals were speech mixtures, which naturally guided LNuDFT to allocate higher resolution to the mid-frequency region where both speech energy and IPD reliability are the strongest. For other acoustic domains (e.g., sound event localization of bird chirps or ambient wind noise), the learned sampling pattern may indeed differ, as LNuDFT adapts to the spectral statistics of the target domain.
>   Moreover, microphone spacing and the sampling rate impose physics-based limits on the range of observable IPD cues (e.g., low-frequency phase ambiguity and high-frequency spatial aliasing). These physical constraints inherently shape the optimal allocation of LNuDFT parameters. We note that exploring how LNuDFT adapts across different acoustic environments or source characteristics would be an interesting direction for future work, as the optimal frequency-bin allocation may vary substantially depending on the underlying signal and array properties.
>
> * rMPE.
>
>   To justify the physics-informed nature of rMPE, we added Appendix A.1 in the revised manuscript. There, we have shown that the TDOA and its corresponding IPD between a pair of microphones depend solely on their relative position vector under the plane-wave model, demonstrating that rMPE explicitly encodes physically grounded geometric relations.
>
> ---
>
> **W2. Difficulty in result analysis: dynamic microphones datasets and real vs. synthetic trends.**
>
> * The goal of our study is to assess the generalization capability of the proposed methods across diverse microphone array geometries. For this purpose, we constructed the synthetic datasets using dynamic microphone arrays, which provide a wide variety of spatial configurations and allow controlled evaluation of geometry generalization.
>
>   As shown in Table 3 of the revised manuscript, our analysis indicates that the primary factor behind performance differences was whether the number of channels at test time matches those seen during training, rather than the dynamic nature of the microphones themselves. Therefore, we explicitly separated the datasets into _Dynamic-S_ (seen channel counts) and _Dynamic-U_ (unseen channel counts) to isolate the effect of channel variability on generalization performance. This clarification is now included in Lines 371–372 of the revised manuscript.
>
> * Regarding Table 4 of the revised manuscript, which evaluated different numbers of grid points $D$, the performance generally improved as $D$ increased and stabilized around $D \geq 512$. For the synthetic datasets, larger $D$ provided slightly better results because finer angular discretization reduced quantization errors under simulated (i.e., matched) conditions.
>
>   In contrast, the real datasets showed saturation or mild degradation beyond $D= 2048$. We attribute this to the fact that overly dense candidate grids made the localization decision more sensitive to small acoustic mismatches inevitably present in real recordings, such as sensor noise, array manufacturing tolerances, and synthetic RIR deviations. These observations, interpretations, and the corresponding conclusions have been added in Lines 475–487 of the revised manuscript.
>
> ---

---

> ### Author Response · Authors · 2025-11-20
> **Response to Reviewer aRq1 (2/2)**
>
> ---
>
> **Q1. Experiment (v) still outperforms the proposed one on the _Dynamic-U_ data and performs on par with the proposed one for _NAO robot_, but significantly degrades on the _Eigenmike_. Please expand on this discrepancy?**
>
> * Experiment (v) used the same logit-based initialization as the proposed method but did not update the LNuDFT parameters during training. Thus, its frequency-bin allocation remained fixed and could not adapt to unseen microphone geometries or acoustic conditions.
> * Synthetic domain (_Dynamic-U_).
>
>   Although _Dynamic-U_ contained unseen channel counts, its overall acoustic conditions were still highly consistent with the simulated training domain (i.e., identical RIR generation process). Under such controlled settings, even a fixed initialization remained effective, which explained why experiment (v) occasionally performed on par with — or slightly better than — the learned version.
> * Real domain (_Eigenmike_).
>
>   The Eigenmike recordings differed substantially from the training conditions in several aspects:
>
>   1. a different number of channels, with a relatively large 32-channel microphone array,
>   2. real-world factors such as sensor mismatch, microphone manufacturing tolerances, and non-ideal RIRs.
>
>   These mismatches degraded the performance of experiment (v), which could not adapt its frequency allocation to compensate for these discrepancies. In contrast, the proposed method learned and adjusted the LNuDFT parameters during training, allowing the model to refine its initial allocation and generalize more effectively to unseen geometries in real-world acoustics.
>
> ---
>
> **Q2. In L70, “Both components facilitate the extraction of spatial representations with physics-based inductive biases”: Why are LNuDFT and rMPE considered as imposing (physics-based) inductive biases? To me, these are perceived as a process that introduces and makes learnable parameters for feature extraction more flexible, and I see it as a process that relieves the inductive bias.**
>
> * We intended to convey that LNuDFT and rMPE incorporate physics-based principles—Fourier analysis and inter-microphone phase relationships—into the feature extraction process. These domain principles guide the model toward acoustically meaningful spatial cues and thus serve as physics-informed inductive biases that improve data efficiency and generalization.
>
>   At the same time, certain parameters of LNuDFT are made learnable, which introduces flexibility and allows the model to adapt to the statistics of the training data. This design balances inductive structure with learnable adaptability, rather than removing inductive bias entirely.
>
>   To avoid misunderstanding, we revised the wording in Lines 70–73 of the manuscript as follows:
>
>   “Both components incorporate physics-based structural assumptions into the feature extraction process while still allowing adaptivity through training. This yields physics-informed inductive biases that guide learning toward acoustically meaningful representations.”
>
> ---
>
> **Q3. Can the LNuDFT initialization or update scheme get stuck in poor local minima, e.g., if the initial frequency allocation is far from optimal? Have the authors tried more physically motivated or data-driven initializations beyond logit-based mapping?**
>
> * LNuDFT, like other gradient-based models, is in principle susceptible to local minima. In our experiments, however, both uniform initialization (equivalent to the standard DFT) and the proposed logit-based initialization consistently converged to similar solutions, allocating denser frequency bins in the mid-frequency range. This indicates that the optimization was relatively stable under the conditions we tested. We agree that extremely poor initialization may lead to suboptimal convergence, and a more systematic analysis of the optimization landscape, as well as the development of robust training strategies, would be an important direction for future work.
> * Regarding alternative initializations, we experimented with:
>
>   1. Uniform initialization, which reliably converged to mid-frequency–dense allocations (Appendix A.2), motivating the use of a physics-informed prior via logit-based initialization,
>   2. Warped DFT initialization [1], which can emphasize low-frequency regions with a nonlinear frequency warping. However, these low-frequency–focused allocations did not improve performance, as mid-frequency IPD cues were more informative for SSL in our target domain.
>
>   Therefore, among the examined schemes, the logit-based initialization provided the most effective inductive prior for learning LNuDFT parameters. Future work could explore more sophisticated data-driven or physics-informed initialization schemes to further enhance convergence and performance.
>
>
>   [1] A Makur and S K Mitra. Warped discrete-Fourier transform: Theory and applications. IEEE Transactions on Circuits and Systems I: Fundamental Theory and Applications, vol. 48, no. 9, pp. 1086-1093, 2001.
>
> ---

---

### Author Response · Authors · 2025-12-02
**Final Remarks to the Reviewers and Area Chairs**

Dear Reviewers and Area Chairs,

We sincerely thank you for the time and effort devoted to evaluating our submission. Reviewers' comments helped us substantially improve the clarity and rigor of the work. Throughout the revised manuscript and the rebuttal, we provided all requested clarifications, added new analyses, and conducted additional experiments, with the key improvements reflected in the updated paper.

We summarize below the main contributions and how each reviewer concern was resolved.

---
## 1. Paper Summary
This paper introduces **audio-geometry-grid representation learning (AGG-RL)**, a physics-informed approach for sound source localization (SSL) that generalizes across a wide range of **microphone arrays** and **arbitrary DOA grids** without retraining.
### Key contributions:
### **Physics-informed components**
- **LNuDFT**: a learnable non-uniform DFT emphasizing mid-frequency regions that carry reliable IPD cues.
- **rMPE**: a relative microphone positional encoding directly adhering to the IPD formulation.
### **Geometry-invariant and grid-flexible SSL pipeline**
- Audio and geometry are jointly encoded into **Audio-Geometry Representations (AGRs)** through LNuDFT and rMPE.
- Arbitrary DOA candidates are represented as **Grid Representations (GRs)**.
- AGRs and GRs are aligned in a shared latent space, enabling retraining-free, flexible SSL.

Across various datasets, AGG-RL consistently outperformed baselines, particularly under **unseen geometries**, and **real-world conditions**, while demonstrating strong generalization to **arbitrary grid configurations**.

Given its ability to handle arbitrary geometries and to adapt to different grid configurations, the approach is broadly applicable to practical scenarios such as robotics and smart devices.

---
## 2. Reviewer Feedback and Our Resolutions
We group the reviewers' main concerns into four themes and summarize how each was fully resolved. Importantly, none of the concerns reflected methodological flaws or inconsistencies; rather, they were clarification- and analysis-oriented, all of which we addressed thoroughly.

---
### (1) Why rMPE and LNuDFT are physics-informed
- LNuDFT allocates denser mid-frequency resolution where IPD cues are most reliable, reflecting spatial aliasing constraints.
- rMPE encodes relative microphone displacement vectors, directly mirroring plane-wave TDOA relationships under the far-field model. These clarifications have been added to the manuscript, resolving earlier conceptual questions.

---
### (2) Novelty, hyperparameters, and initialization of LNuDFT
- We clarified that the novelty of LNuDFT lies in its learnable frequency sampling tailored to the SSL task, unlike fixed approach. We also introduced a logit-based approach that emphasizes mid-frequency regions from the start. The hyperparameters of LNuDFT were selected to respect non-aliased regions and clipped within a bounded range to ensure monotonicity and compliance with the Nyquist limit.
- We considered the initialization with warped-DFT and uniform strategies. Among these, the logit-based approach—which was designed to emphasize mid-frequency regions—yielded the most stable convergence.

---
### (3) Dataset-specific trends and ablation analyses
- We clarified the different performance trends observed in the ablation studies between synthetic and real datasets, as well as between seen and unseen channel configurations. These trends primarily stem from the contrasting characteristics inherent to each dataset. As a result, fixed NuDFT and fixed grid ablation methods interact differently with each dataset’s characteristics.
- We also explained why the effect of grid resolution varies across datasets. In synthetic settings, finer grids consistently improved performance due to their matched acoustic conditions. In real-world datasets, however, the advantages were less pronounced because mismatches were present, limiting the benefits of finer grids. These clarifications, added to the revised manuscript, strengthen the technical interpretation of the results.

---
### (4) Additional experiments and detailed breakdowns
In response to reviewer requests, the revised manuscript now includes:

- Performance breakdown by SNR, RT60, and number of speakers
- MAE spread via 95% confidence intervals
- Additional cross-dataset experiments on _STARSS23_

These analyses provide a more comprehensive evaluation and further validate the robustness of AGG-RL across diverse scenarios.

---
## Final Remark
All concerns raised during the review process were fully resolved through additional analyses, expanded experiments, and clear technical explanations. We also note that, in the discussion, a reviewer acknowledged that the questions were satisfactorily addressed.

We believe that this study offers a novel, robust, and practically valuable contribution to SSL generalization across diverse geometries and grids. We respectfully request favorable consideration.

---

### Meta-Review · Area_Chair_dpQm · 2025-12-28

**Summary:**

The paper introduces AGG-RL, a framework for sound source localization (SSL) that aims to solve the problem of retraining-free generalization across varying microphone array geometries and arbitrary direction-of-arrival (DOA) grids. I recommend accepting this work because it provides a genuinely elegant and practical solution to geometry invariance by jointly encoding audio, geometry, and grids into a shared latent space. The authors' ability to handle diverse arrays, from robot microphones to 32-channel arrays and without retraining, represents a valuable contribution to real-world acoustic sensing.

**Reviewer Concerns:**

Reviewer aRq1 was primarily concerned with the lack of explicit justification for the physics-informed claims regarding the LNuDFT and rMPE components. Reviewer LRbg focused on the limited scope of the initial evaluation, specifically requesting breakdowns by SNR, reverberation, and speaker counts, as well as broader cross-dataset validation. Reviewer ZXmA questioned the stability and initialization of the learnable frequency sampling and noted performance trade-offs between flexible and fixed-grid configurations in synthetic data. Most concerns were addressed in the rebuttal and some reviewers noted that future work could further explore how LNuDFT adapts to non-speech signals (e.g., bird chirps).

**Reviewer Scores:**

Based on the above issues, I think Reviewer LRbg will increase to 6, the other two will keep their scores, finally 6,6,6.

---

### Decision · Program_Chairs · 2026-01-26

Accept (Poster)